# Research on quantitative evaluation of China's Intelligent Construction Policy (CICP) based on the integration of PMC index model and multi-dimensional analytical framework

Xiongquan Ou[1], Ming Ma[2,3]*, Wei Wang[4]

**1** School of Art Design and Media, Sanda University, Shanghai, P.R.China, **2** College of Architecture and Urban Planning, Yunnan University, Kunming, Yunnan, P.R.China, **3** Institute for Smart City of Chongqing University in Liyang, Changzhou, Jiangsu, P.R.China, **4** School of Architecture and Planning, Hunan University, Changsha, Hunan, P.R.China

* mming@ynu.edu.cn

## Abstract

This study presents a systematic evaluation of China's Intelligent Construction Policies (CICP) from 2010 to 2022, employing a quantitative approach based on the Policy Modeling Consistency (PMC) index model. Through text mining and bibliometric analysis, we assess the formulation quality, evolution, and effectiveness of 30 national policies, identifying distinct phases of development: the Cultivation and Exploration Phase (CEP: 2010–2019) and the Development and Promotion Phase (DPP: 2020–2022). The results demonstrate a significant improvement in policy quality, with the DPP achieving higher PMC scores and more frequent issuance of "Excellent" and "Positive" policies. Key findings reveal the transition from fragmented, exploratory policies to integrated, goal-oriented strategies, emphasizing industrialization, digitalization, and sustainability. However, gaps remain in policy scope, effectiveness, and incentive mechanisms. The study concludes with targeted recommendations for optimizing CICP, including expanding policy scope, enhancing implementation frameworks, and fostering public-private innovation ecosystems to align with technological advancements and industry demands.

## Introduction

The adoption of emerging technologies to address traditional construction challenges has become a national development strategy in China [1]. Intelligent construction, which integrates the latest information technology with construction engineering [2], relies on core technologies such as "Artificial Intelligence" (AI), "Building Information Modeling" (BIM), "Big Data," "Cloud Computing," "Blockchain," "Internet of Things (IoT)," and "Robotics" [3]. These core technologies significantly enhance construction efficiency and precision. AI optimizes decision-making and project management, BIM

**Data availability statement:** All relevant data are within the paper and its Supporting information files. All of the dataset files are also available from the figshare database (https://doi.org/10.6084/m9.figshare.29116007).

**Funding:** This study was supported by [Open Program of the Key Laboratory of Mountain Town Construction and New Technologies of the Ministry of Education] in the form of a grant awarded to [M. Ma] (LNTCCMA-20250106) and [Yunnan University]; [General Program of the Basic Research Special Program of the Department of Science and Technology of Yunnan Province] in the form of a grant awarded to [M. Ma] (202401CF070194) and [Yunnan University] in the form of a salary for M. Ma. The specific roles of this author are articulated in the 'author contributions' section. The funders had no role in study design, data collection and analysis, decision to publish, or preparation of the manuscript.

**Competing interests:** The authors have declared that no competing interests exist.

improves collaboration and planning, while Big Data and Cloud Computing enable real-time monitoring and resource optimization. Blockchain ensures secure and transparent transactions, IoT facilitates automation through connected systems, and Robotics boosts construction speed and safety. Collectively, they drive the shift from digitalization and automation to integrated, intelligent construction, achieving higher efficiency and energy savings [4]. In recent years, with the acceleration of global digitalization, fabricating powerhouses like the USA, Japan, and European countries have enacted a series of policies to promote the industrialization of intelligent construction. As the world's largest construction market, China plans to keep pace with this trend and accelerate the development of intelligent construction.

Since 2010, China has issued a series of national policies to promote digitalization and intelligent transformation in the construction industry. Intelligent construction has seen further growth and expansion in China's push for new industrialization and new infrastructure policies [5] issued by the central government from 2020 to 2022. As 2023 began, there was a decrease in the central government's issuance of national-level policies on intelligent construction. Meanwhile, there was a significant increase in the release of local-level policies, reflecting differences in regional development strategies. Consequently, the era from 2010 to 2022 marks a critical period in developing China's intelligent construction policies (CICP) at the national level. However, the in-depth promotion of CICP faces several challenges. These include imperfections in policy guarantees and a lack of comprehensive content in the policies.

Research on CICP remains notably limited, with existing studies primarily relying on empirical feedback or qualitative analysis. A critical research gap persists in the absence of rigorous quantitative evaluations of policy texts, significantly constraining the capacity to assess policy effectiveness with scientific precision. This gap matters because a quantitative approach would improve the accuracy and objectivity of policy assessments, allowing for a more systematic evaluation of the strengths and weaknesses of policies. Without such a framework, it is difficult to quantify the actual impacts of these policies or identify areas for improvement. Therefore, it is crucial to explore data-driven methodologies that can assess policy texts quantitatively. Key considerations include developing a robust evaluation system and determining the quality of content within existing policy texts, ensuring that the evaluation is both comprehensive and scientifically grounded. This paper aims to provide a scientific analysis and evaluation of the CICP based on a suitable methodology.

Effective policy implementation is contingent upon the quality of policy formulation, and the evaluation of policy texts forms a critical foundation for policy-making [6]. Text mining, in conjunction with bibliometric analysis, represents a methodical approach to quantitatively examining policy texts. Policy texts can effectively employ bibliometric methodologies developed for academic literature, facilitating rigorous quantitative analysis, and this approach has been substantiated as viable in numerous scholarly endeavors [7]. The Policy Modeling Consistency (PMC) index model [8], rooted in sophisticated text mining methods, provides a comprehensive understanding of policy texts, encompassing their structure, content, and consistency. As a distinct

metric, the PMC index model not only intuitively assesses a policy's strengths and weaknesses via a scoring system but also clarifies the contribution of each variable to the overall score, ensuring that policies are effective and aligned with the industry's evolving needs. Consequently, this paper employs text mining, bibliometric analysis, and the PMC index model to quantitatively analyze and scientifically evaluate the CICP, offering relevant insights and recommendations. Based on the current research status, research objectives, and significance of CICP evaluation, the research addresses the following key areas:

Firstly, this paper evaluates the formulation quality of CICP based on the PMC index model and defines the phases of policy development.

Secondly, this paper reviews and discusses the primary features of CICP from a dynamic analysis framework, thinking about the causal relationships behind policy formulation and development in phases.

Finally, this paper discusses the future trajectory of CICP, finding potential areas for optimization based on PMC surfaces and emphasizing the research frontiers in intelligent construction to identify significant gaps in the current policy framework.

The rest of the paper is structured as follows: the second part reviews the related literature; the third part introduces the methodology and detailed research design of this paper; the fourth part presents the results and discussion; the fifth part is the summary and conclusion. The sixth part discusses the limitations.

## Literature review

### Research on CICP

(1) Themes of the existing studies

Research on CICP currently spans multiple dimensions, including technology, industry, education, and management, aims to provide actionable recommendations. The existing studies can be grouped into the following thematic areas:

**Technology application and infrastructure construction.** Liu et al. [3] analyzed policies supporting core technologies in intelligent construction, proposing measures such as advancing infrastructure investment, promoting technology application demonstrations, strengthening public technology provision, and providing financial and tax support.

**Environment improvement and platform support.** Chen and Ding [9] emphasized the influence of market environments and resource platforms on the development of intelligent construction. Their recommendations included establishing innovation bases, fostering R&D cooperation, and promoting collaborative training between universities and enterprises.

**Standard-setting and management innovation.** Wang [10] reviewed domestic and international policies on intelligent construction, suggesting the improvement of technical standards, encouragement of collaborative growth, and management innovation.

**Talent development and system optimization.** Li [11] summarized key measures in CICP, such as fostering collaborative advancement, advancing talent training, cultivating industrial systems, innovating service models, increasing pilot projects, and improving service systems.

(2) Gaps in current research

As summarized in the previous section, Current research on CICP mainly focuses on suggestions from scholars and experts, typically in the form of qualitative analysis or experiential judgment. While expert-based policy suggestions reflect professionalism and foresight, ensuring their comprehensiveness and systematic evaluation remains a challenge. The existing studies highlight the importance of improving policy and institutional frameworks, but most of the recommendations lack empirical evidence. As a result, there is a notable gap in the scientific research, particularly in terms of quantitative analysis of CICP.

(3) Advances in quantitative policy analysis

Several studies have advanced in applying quantitative methods to policy analysis in related fields. For instance:

Zhang et al. [12] developed a "theme-instrument-evaluation" framework to analyze digital transformation policies in China's construction industry.

Li [13] proposed a "policy phase-policy tool-text type" framework to study the evolution of China's prefabricated building policies, offering insights for policy implementation.

Peng et al. [14] created a "policy phase-policy tool-policy goal" framework to examine the evolution of urban circle policies in China, highlighting the multi-dimensional coupling characteristics.

These studies offer valuable methodological insights for this paper's quantitative analysis of CICP. By developing a multi-dimensional analytical framework, this research aims to quantitatively assess CICP, identify policy features, and understand the outcomes of policy formulation.

## Policy text quantitative analysis

In recent years, the approach to analyzing policy texts has shifted from qualitative to quantitative ones, focusing on a combination of descriptive implementation and statistical validation to facilitate scientific policy formulation and decision-making [15]. As a unique type of document, policy texts have adopted and innovated upon the theoretical methods of bibliometrics from the academic literature due to their structural similarities [7]. The scientific nature of this approach lies in its use of extensive data analysis to extend the scope and coverage of research. However, based on large samples, this method also tends to overlook individual characteristics in pursuing standard textual features [7], often necessitating the integration of content analysis in the research process.

Several research perspectives exist, such as policy evolution, tools, and evaluation. A multi-tiered methodological approach [15] has been formed: (1) Utilizing data statistics to analyze characteristics such as text publication time, keywords, and issuing institutions; (2) interpreting the content of texts from aspects such as policy themes, goals, and tools, combined with qualitative analysis for a logical explanation; (3) delving deeper into policy information and its connotations for a comprehensive evaluation and recommendations. With societal development, new research methods have emerged with the rise of big data technologies, such as examining the interrelationships between policy-making entities from a network perspective and enhancing understanding of text semantics using knowledge graphs. The differences in samples, algorithms, and tools lead to varied research outcomes. An appropriate analysis framework needs to be constructed based on the attributes of the research objectives.

## Research on policy evaluation

Policy evaluation is a feedback mechanism for refining existing policies and a predictive tool for shaping new ones [16], guiding their formulation, adjustment, and optimization [17]. This process requires a comprehensive review of the policy system, utilizing scientific standards and methods to judge and measure policy texts [17,18,19].

In the early times, policy evaluation was predominantly empirical. Since the 1970s, policy evaluation has evolved from empirical to normative standards, focusing on value judgments. This shift is exemplified by Suchman's five evaluation types [20] and Poland's "three E" framework [21]. As research progresses, policy evaluation now integrates content analysis with expert insights. It concentrated on semantic analysis, primarily through content analysis, by comparing high-frequency and thematic words in policy texts [22,23]. While qualitative evaluation has merits in analysis and summarization, its subjectivity regarding evaluation indicators and lack of objective reference points has been criticized [24].

Consequently, there is a growing trend of incorporating quantitative approaches in policy evaluation. Beginning as early as 1978, Libecap [25] compared current and past policies to create a legal change index for evaluating Nevada's mining laws by sorting out the scores of all policies throughout the year. After that, scholars worldwide have researched

quantitative analysis in policy evaluation, most relying on scoring, and the topics cover finance, science, technology, healthcare, and the environment. [23,26–28]

In contemporary policy evaluation, the focus shifts from exclusively qualitative or quantitative methods to integrating both [29], blending qualitative approaches like case studies and expert reviews with quantitative analyses using mathematical models. Empirical approaches such as text data mining [30,31], social network analysis [32], and fuzzy comprehensive evaluation [33] have become comprehensive. Standard empirical tools include PSM-DID model analysis [34], instrumental variables [35], and synthetic control [36]. Advancements in quantitative analysis tools for policy texts also show new progress. Natural language processing (NLP) tools convert text into numerical data, enabling the analysis of semantic information [37,38]and extraction of high-frequency vocabulary, thus distilling key policy messages [39]. This approach, now prevalent in policy text analysis [40–43], typically employs software like Python [44], Rost Content Mining 6.0 [45], and ROSTCM 6.0 [46,47]. Furthermore, the PMC Index model has been introduced to evaluate individual policies and facilitate comparative analysis across multiple policies, marking a significant stride in policy evaluation.

### Application of the PMC index model

The PMC index model was established by Ruiz Estrada et al. [8,48]and is based on the Omnia Mobilis hypothesis [49] This hypothesis suggests that everything is interconnected and dynamic, urging a comprehensive approach to assessing the consistency of policy texts. The PMC index model merges Cartesian spatial application with text mining, offering a multi-dimensional evaluation that visualizes policy texts' strengths and weaknesses. Scholars have applied it to evaluate digital transformation policies [12], land protection policies [50], big data development policies [51], insurance policies [52,53], pork industry policies [46], disaster relief policies [54], ecological protection compensation policies [47], and emergency response policies [38] evaluation. Zhang et al. [55–59] applied it to innovation policies, financial policies, new energy vehicle subsidies policies, and more, contributing significantly to China's policy evaluation framework. Furthermore, based on the PMC index model, Zang et al. [60] quantitatively analyzed China's AI policies and proposed policy formulation suggestions in line with research frontiers, and Hu et al. [61] quantitatively evaluated China's robot industry policies, providing a decision-making basis for policy optimization. These applications demonstrate the feasibility and value of using the PMC index model for evaluating CICP.

### Contributions of this research

Given the lack of application of quantitative analysis methods in CICP, the PMC Index model is introduced into the analysis process in this research, and a multi-dimensional discussion of the policy features and formulation is conducted. Furthermore, considering the policy's timely and ever-evolving nature, this research integrates a dynamic analysis of temporal changes to compare the CICP at various phases. The approach facilitates a more comprehensive understanding of the CICP's evolution and status features. As a result, it allows for a more nuanced exploration of policy trends. The systematic and dynamic methodology in this research offers a fresh perspective for policy analysis and evaluation, moving beyond conventional static methods.

### Methods and materials

#### Methodological framework

Based on references [13,14,62–64], this research constructed an advanced multi-dimensional analytical framework that applies text mining and data visualization analysis techniques to rigorously analysis the CICP. The framework covered evaluation, discussion, and optimization recommendations for CICP, and be executed according to five parts of advanced work (see Fig 1), including the policy evaluation based on the PMC Index Model and the dynamic analysis of policy evolution, status, and trend, offering data-driven recommendations for the optimization of CICP.

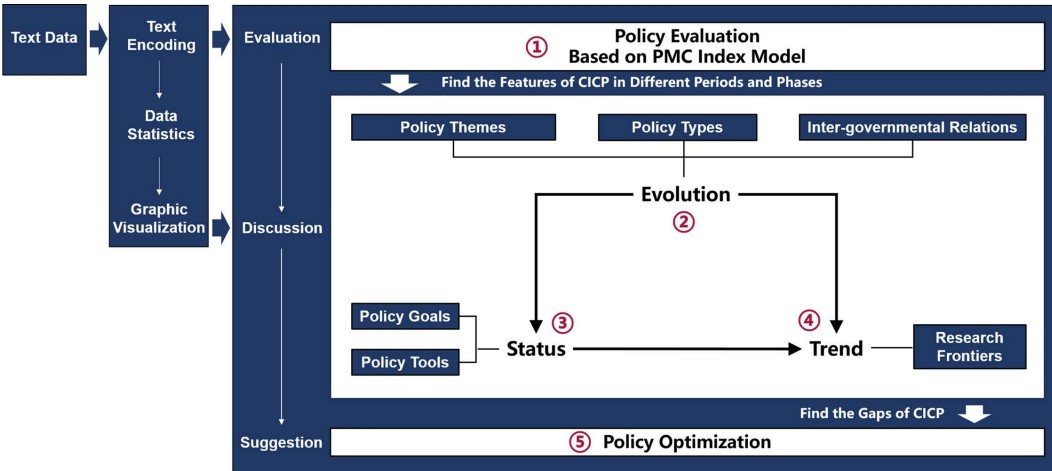

**Fig 1. Analytical framework.**

## Research method and work procedure

This study employed methods such as text encoding, data analysis, and graphical visualization. The research process consisted of five stages: data acquisition, data processing, computation and analysis, results discussion, and conclusion. (see Fig 2) After the computation and visualization of the data, the five parts of advanced work constructed in the analysis framework above (see Fig 1) were carried out, and the corresponding findings and prospects were proposed. The following is an introduction to the research methods and working procedures according to contents established by the framework, these are also described in detail in Fig 2.

(1) Evaluate the formulation of CICP based on the PMC index model

Firstly, the relevant texts of CICP were collected and collated based on the information disclosure platform, and a research database was established. Then, the PMC index model was used to scientifically analyze and evaluate the formulation of CICP, and then the phases of CICP's development were summarized and discussed. In this process, the evaluation of CICP using the PMC Index Model constitutes the most critical section. As delineated in the preceding literature review, the PMC index model enables quantitative assessment of policy comprehensiveness and coherence through the establishment and scoring of multidimensional indicators. This method facilitates the identification of strengths and weaknesses in policy formulation through data-driven analysis, thereby providing empirically grounded insights for future policy optimization. The subsequent section will elaborate on the application methodology of this model with specific exemplifications.

The evaluation procedure follows a standard four-step process for calculating the PMC Index [8], combined with content analysis of policy texts. The steps are outlined as follows:

### Step 1: Define computational variables and identify parameters

This step involves setting up primary and secondary variables based on the selected policies within the sample. Each variable is assigned specific parameters, which are configured to align with the text content and intended policy goals. In general practice, primary variables are employed to assess the consistency of policies. These variables exhibit minimal variation in their configuration across different studies, allowing researchers to reference classical settings from existing applications of this model. By referring to the classic research methodology of authoritative scholars and combining it with the development demands of China's intelligent construction, the selection and setting of primary variables can be carried

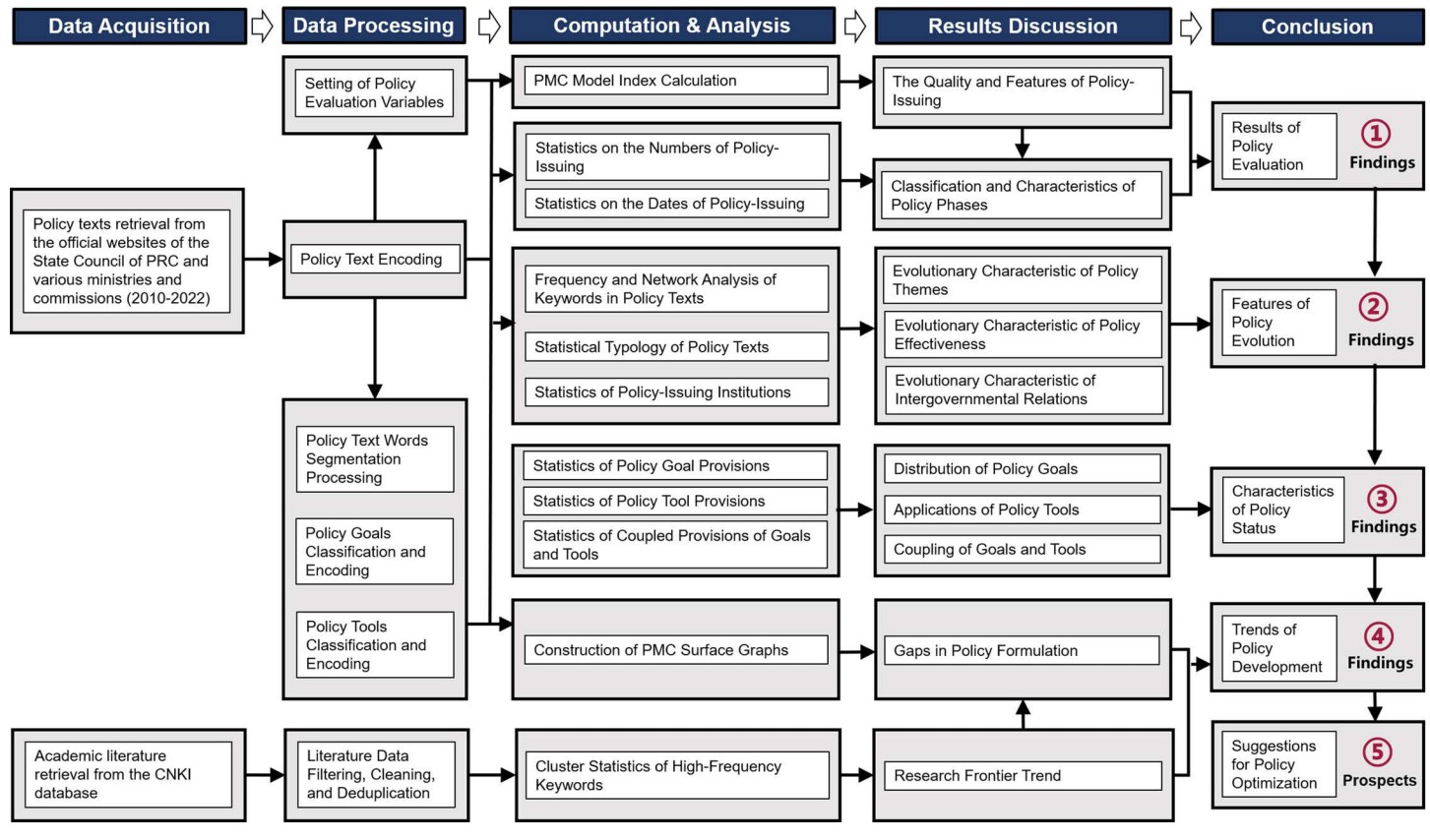

**Fig 2. Research method and work procedure.**

out. Secondary variables, operating within the scope of primary variables, facilitate a more granular analysis and comparison of various dimensions and factors in policy development. These variables are particularly instrumental in identifying deficiencies and shortcomings in policy formulation. Consequently, their determination requires customization based on the specific characteristics of the analytical subject. The selection of corresponding variables typically derives from a systematic synthesis of research findings within the subject's domain, complemented by empirical feedback from practical implementation experiences.

For example: the policy text (e.g., 'Policy P0') is evaluated within a standardized framework of ten primary variables, including "X1 Policy Nature," "X2 Policy Effectiveness," and so forth. Within the scope of these primary variables, a hierarchical system of secondary variables is systematically developed. To illustrate, X1 incorporates two secondary variables: "X1:1 Prediction" and "X1:2 Regulation." Similarly, X2 is structured with three secondary variables: "X2:1 Short-term," "X2:2 Medium-term," and "X2:3 Long-term," among others. It is noteworthy that certain primary variables, such as X10, may remain without secondary variables based on analytical requirements. (see Table 1)

### Step 2: Conduct calculations using the PMC index formula

In this step, calculations are carried out using the classical PMC Index formula. Through a comprehensive content analysis of the policy text, the secondary variables of the target policy can be quantified by referencing Formula (1) and Formula (2). This is done by evaluating the presence or absence of specific variables in the policy texts and assigning binary values (see Table 1).

**Table 1. Evaluation index system and criteria for P0.**

| Primary Variable | Secondary Variable | | | | Evaluation Criteria |
|---|---|---|---|---|---|
| X1 Policy Scope | X1:1 Prediction | X1:2 Regulation | | … | Score 1 for fitting each secondary variable description; otherwise, score 0. |
| X2 Policy Effectiveness | X2:1 Short-term | X2:2 Medium-term | X2:3 Long-term | … | |
| … | | | | … | |
| X10 Policy Disclosure | | | | … | |

$$X \sim N = [0, 1] \tag{1}$$

$$X = \{XR : [0 \sim 1]\} \tag{2}$$

Secondary variables are treated in binary form, with equal weighting applied. Once secondary variables are processed, primary variables are calculated using Formula (3), and the overall PMC Index is computed using Formula (4). The aggregate score of a policy formulation is derived through the summation of ratings across all primary variables.

$$X_t \left( \sum_{j=1}^{n} \frac{X_{tj}}{T(X_{tj})} \right) t = 1, 2, 3, 4, 5 \cdots \infty \tag{3}$$

$$PMC = \begin{bmatrix} X_1 \left( \sum_{i=1}^{5} \frac{X_{1i}}{5} \right) + X_2 \left( \sum_{i=1}^{3} \frac{X_{2i}}{3} \right) + X_3 \left( \sum_{i=1}^{4} \frac{X_{3i}}{4} \right) + \\ X_4 \left( \sum_{i=1}^{3} \frac{X_{4i}}{3} \right) + X_5 \left( \sum_{i=1}^{2} \frac{X_{5i}}{2} \right) + X_6 \left( \sum_{i=1}^{4} \frac{X_{6i}}{4} \right) + \\ X_7 \left( \sum_{i=1}^{5} \frac{X_{7i}}{5} \right) + X_8 \left( \sum_{i=1}^{5} \frac{X_{8i}}{5} \right) + X_9 \end{bmatrix} \tag{4}$$

(i = primary variable; j = secondary variable; n is the total number of secondary variables.)

For example Generally, a multi-input-output table is used to handle extensive data for each variable, serving as the foundation for analyzing primary variables. Binary values are assigned to secondary variables, and these are used to calculate the primary variables (see Table 2). For instance, during the textual analysis of Policy P0, the scoring process was conducted as follows: Regarding the "X1 Policy Scope" variable, we classified the policy scope as "X1:1 Prediction" rather than "X1:2 Regulation," resulting in a score of 0.50 for X1. For the "X2 Policy Effectiveness" variable, we determined that the policy's efficacy primarily targets short-term and medium-term outcomes rather than long-term effects, yielding a score of 0.67 for X2. This scoring methodology was consistently applied across subsequent variables. Specifically, for X10 "Policy Disclosure," we assessed the policy as fully transparent, assigning a maximum score of 1.00. Through the aggregation of scores from all ten primary variables, we obtained a composite score of 7.50, which represents the final evaluation of P0 policy formulation based on the PMC index model.

**Table 2. Multi-input–output table for P0 (for example).**

| X1 (score:0.50) | | X2 (score:0.67) | | |
|---|---|---|---|---|
| X1:1 | X1:2 | X2:1 | X2:2 | X2:3 |
| …. | | | | |
| …. | | | | |
| X10 (score:1.00) | | | | |

**Step 3: Policy rating based on PMC index**

The composite score can be subsequently categorized according to predefined grading standards, facilitating both hierarchical classification and comparative analysis among different policies. The rating criteria are set as follows in normal: [0, 5.99] is Average, [6, 6.99] is Acceptable, [7, 7.99] is Positive, [8, 8.99] is Excellent, and [9,10] is Perfect. Such an analytical framework also enables the identification of relative strengths and weaknesses, thereby providing a robust basis for policy evaluation and improvement.

For example Based on the computational results derived from the PMC model, Policy P0 attained a score of 7.50, which, according to the established evaluation criteria, categorizes its formulation quality within the "Positive" classification tier. This methodological framework can be systematically replicated to evaluate and score other policies, thereby enabling comparative analysis and cross-policy benchmarking.

**Step 4: Construct PMC surfaces based on PMC index**

PMC surface drawing is employed to facilitate multi-dimensional policy evaluation and allow for a deeper single-indicator analysis. This method is especially useful for assessing the internal consistency of policies and comparing various approaches. The PMC surfaces are calculated using third-order matrices as Formula (5), which provide a visual depiction of the evaluation outcomes. The variance between the evaluated policy and an ideal "perfect" policy is highlighted, which can be quantified using the concavity index [59]. The concavity index is calculated as the perfect score of 10 minus the variable's PMC evaluation score, and the concavity index composition refers to the proportion of a variable's concavity index relative to the total concavity indices across all variables, thereby enabling the quantification of policy formulation deficiencies within specific variable dimensions.

$$PMCSurface = \begin{pmatrix} X_1 & X_2 & X_3 \\ X_4 & X_5 & X_6 \\ X_7 & X_8 & X_9 \end{pmatrix}$$

(5)

For example Based on the evaluation scores of P0's nine primary variables (note that X10 is exclusively designated for assessing policy transparency and thus excluded from this analytical phase), a three-dimensional matrix is constructed as specified in Formula (5). This matrix serves as the computational foundation for generating corresponding surface diagrams using analytical software such as EXCEL. Through integrated analysis combining graphical representations with the composition of primary variables' concavity index (see Fig 3), we can systematically identify and quantify the shortcomings in the policy formulation process of P0. The graphical analysis reveals that Policy P0 demonstrates optimal formulation in the dimensions of X3, X4, and X7, as evidenced by their respective scores and graphical representations. However, deficiencies are identified in the X1, X6, and X9 dimensions, indicating areas requiring substantial improvement and refinement in the policy formulation process.

**(2) Discuss the evolution of CICP**

Following the quantitative evaluation of CICP's formulation quality and its developmental stage classification, a comprehensive investigation into the distinctive features of CICP's evolution is warranted. This subsequent analysis aims to delve deeper into the intrinsic patterns and unique attributes that have shaped CICP's developmental trajectory, thereby providing valuable insights for future policy refinement and strategic planning. The dimensions of analysis include policy themes, policy types, and intergovernmental relations, focusing on how policy supply and demand drive these changes examining government decision-making and governance shifts. The NLP tool, ROSTCM6 from Wuhan University, provides text analysis methods, including word segmentation and frequency assessment. It helps extract keywords and create semantic networks, revealing the main themes and hotpots in policy discussions. Additionally, the research assesses policy effectiveness and government collaboration using policy types and issuing institutions' statistics.

 

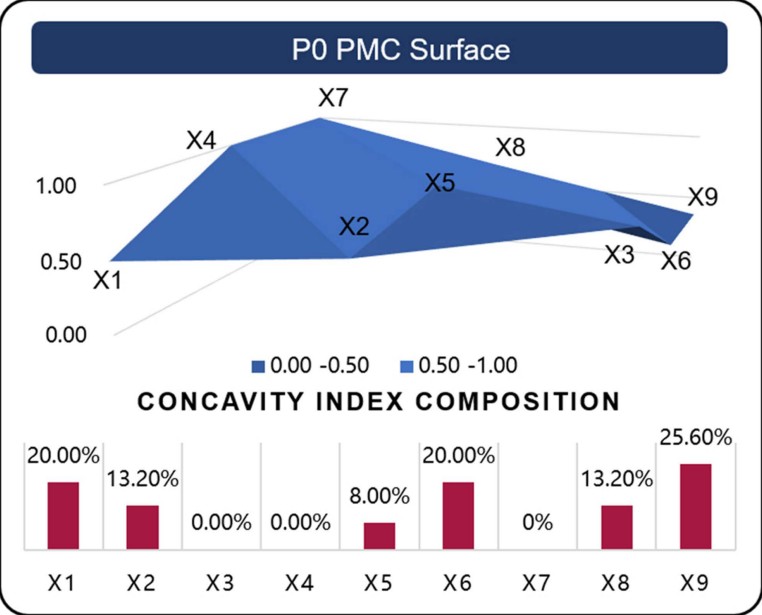

**Fig 3. PMC Surfaces and Concavity Index Distribution for P0 (example).**

**(3) Discuss the status of CICP**

Having identified the evolutionary features of CICP, the analysis proceeds to discuss the characteristics of policy status. The dimensions of analysis include policy goals and tools. Policy tools are interventions policymakers adopt to achieve their goals [65], reflecting the government's policy values and governance concepts [66]. The research considers the structural characteristics of policies and government intervention methods by analyzing the distribution of policy goals and tools in different periods and the coupled application of goals and tools.

**(4) Identify the trend and find the gaps in CICP**

This research utilizes the CNKI database for a three-year literature review, employing the software VOSviewer for bibliometric analysis. This method generates a table of high-frequency keyword clusters essential for identifying key trends in China's intelligent construction field. The aim is to find the policy gaps and discover potential optimization areas of the CICP by integrating the findings of policy evaluation based on the construction of PMC Surfaces.

**(5) Propose the suggestions of CICP**

Based on the findings and discussions above, the paper advances policy recommendations to foster the rapid progression of intelligent construction. These suggestions enhance the comprehensiveness and adaptability of policy formulation in this domain.

**Data sources and sample description**

**Data collection methods.** This research sourced policy texts from China's State Council (SC) and national ministries' websites, ensuring data authenticity and precision. The policy text selection followed these principles:

(1) The keyword "intelligent" was used in the policy text search. The selected texts had to contain policy content related to intelligent construction in the construction industry.

(2) The research was limited to national policies issued by central government institutions, including various document types like plans, notices, programs, opinions, and outlines.

(3) The cutoff for text issuance was December 31, 2022.

**Research sample processing and coding.** This research gathered 30 national policy texts on intelligent construction, adhering to the outlined search process and selection principles. Table 3 presents these texts, coded for statistical analysis. Table 4 demonstrates the encoding rules developed based on the structural characteristics of the policy texts, including further encoding and categorizing the content of the text clauses.

## Results and discussions

### Policy evaluation based on the PMC index model

**Parameter setting.** As delineated in the preceding methodological exposition, the selection of primary variables in this investigation is grounded in established classical configurations derived from extant research. The formulation of secondary variables, conversely, is predicated upon a nuanced consideration of the specific developmental status and exigencies inherent in intelligent construction policies, thereby ensuring the analytical framework's responsiveness to contemporary policy dynamics.

Based on Estrada's theory [48] and incorporating insights from other scholars [57,59,60] in policy evaluation, this study selected ten primary variables to analyze thirty policies within the feature analysis of CICP. The thirty-six secondary variables were carefully selected and configured within the framework of primary variables, incorporating a comprehensive review of intelligent construction policy research in China, complemented by expert consultations and substantiated by practical implementation experiences [3,9–14]. This rigorous selection process ensures the variables' relevance and applicability to the specific context of the CICP landscape. The specific variables and their corresponding parameters are detailed in Table 5, which outlines the evaluation criteria used for the analysis.

"Policy Scope" serves as a fundamental dimension, examining the various forms policies take in promoting intelligent construction development. This includes predictive functions that anticipate industry trends, regulatory mechanisms that establish operational frameworks, advisory roles that provide guidance, supportive measures that facilitate implementation, and guiding principles that shape strategic direction, along with provisions for unique policy characteristics. "Policy Effectiveness" also serves as a crucial dimension, distinguishing between short-term, medium-term, and long-term impacts. This temporal analysis enables the assessment of policy sustainability and phased outcomes, providing insights into the dynamic nature of intelligent construction development.

To ensure comprehensive sectoral coverage, the "Policy Field" addresses the primary areas influenced by CICP, encompassing economic, social, technological, and environmental considerations. This broad perspective is complemented by the "Policy Perspective" analysis, which examines policies across macro, meso, and micro levels of governance, facilitating a thorough understanding of policy impacts at different operational scales.

The evaluation framework further investigates policy implementation through two critical dimensions: "Policy Function" and "Incentives and Constraints". "Policy Function" focuses on the operational mechanisms employed to promote intelligent construction, including government procurement processes, standardization guidance, regulatory frameworks, technological innovation support, and industry development strategies. The "Incentives and Constraints" evaluates the specific policy instruments utilized, such as investment subsidies, legal protections, talent incentives, tax benefits, intellectual property rights, and support services.

To assess the quality of policy formulation, the "Policy Evaluation" establishes rigorous criteria, examining aspects such as research adequacy, objective clarity, planning detail, and scientific rigor in scheme design. This quality assessment is further enhanced by examining the "Issuing Institution", which analyzes the hierarchical level of policy-making bodies, providing insights into the authority and scope of policy decisions. The framework also incorporates "Policy Recipients"

**Table 3. Research sample of national intelligent construction policy texts (2010-2022).**

| ID | Issue Date | Issuing Department | Policy Text Title | Text Type |
|---|---|---|---|---|
| P1 | 2010/10/18 | State Council (SC) | Decision on Accelerating the Cultivation and Development of Strategic Emerging Industries | Decision |
| P2 | 2011/6/23 | National Development and Reform Commission (NDRC); Ministry of Science and Technology(MST); Ministry of Industry and Information Technology (MIIT); Ministry of Commerce (MOFCOM); State Intellectual Property Office (SIPO) | Guidelines on the Priority Development of High-tech Industrialization Key Areas (2011) | Guidelines |
| P3 | 2011/5/10 | Ministry of Housing and Urban-Rural Development (MOHURD) | Outline for the Development of Information Technology in the Construction Industry (2011–2015) | Outline |
| P4 | 2014/3/17 | SC | National New Urbanization Plan (2014–2020) | Plan |
| P5 | 2014/8/27 | NDRC; MIIT; MST; Ministry of Public Security (MPS); Ministry of Finance (MOF); Ministry of Land and Resources (MLR); MOHURD; Ministry of Transport (MOT) | Guidance on Promoting the Healthy Development of Smart Cities in China | Guidance |
| P6 | 2015/5/8 | SC | Made in China 2025 | Plan |
| P7 | 2015/6/16 | MOHURD | Guidance on Promoting the Application of Building Information Modeling | Guidance |
| P8 | 2016/8/23 | MOHURD | Outline for the Development of Information Technology in Construction Industry (2016–2020) | Outline |
| P9 | 2016/9/30 | SC | Guidance on Vigorously Developing Prefabricated Buildings | Guidance |
| P10 | 2016/12/2 | MOHURD | Unified Standards for Building Information Model Application | Standard |
| P11 | 2017/2/24 | SC | Opinions on Promoting the Sustained and Healthy Development of the Construction Industry | Opinion |
| P12 | 2017/4/26 | MOHURD | Construction Industry Development Plan for the 13th Five-Year Period | Plan |
| P13 | 2017/12/19 | SC | Several Opinions on Deepening the Integration of Production and Education | Opinion |
| P14 | 2019/3/27 | MOHURD | Standards for Quality Inspection of Intelligent Building Engineering | Standard |
| P15 | 2020/7/3 | MOHURD; NDRC; MST; MIIT; Ministry of Human Resources and Social Security (MHRSS); Ministry of Environment and Ecology (MEE); MOT; Ministry of Water Resources (MWR); State Administration of Taxation (SAT); State Administration for Market Regulation (SAMR); China Banking and Insurance Regulatory Commission (CBIRC); State Railway Administration (SRA); National Civil Aeronautics Administration (NCAA) | Guidance on Promoting the Coordinated Development of Intelligent Construction and Construction Industrialization | Guidance |
| P16 | 2020/7/15 | MOHURD; NDRC; Ministry of Education (MOE); MIIT; People's Bank of China (PBC); NBFSA; CBIRC | Action Plan for Green Building Creation | Plan |

*(Continued)*

**Table 3.** (Continued)

| ID | Issue Date | Issuing Department | Policy Text Title | Text Type |
|----|-----------|-------------------|------------------|-----------|
| P17 | 2020/8/28 | MOHURD;<br>MOE;<br>MST;<br>MIIT;<br>Ministry of Natural Resources (MNR);<br>MEE;<br>PBC;<br>SAMR;<br>CBIRC | Several Opinions on Accelerating the Development of New Types of Construction Industrialization | Opinion |
| P18 | 2020/9/8 | NDRC;<br>MST;<br>MIIT;<br>MOF | Guidance on Expanding Investment in Strategic Emerging Industries to Nurture New Growth Points and Poles | Guidance |
| P19 | 2021/2/2 | MOHURD | Notice on Agreeing to Carry Out Intelligent Construction Pilot Projects | Notice |
| P20 | 2021/3/16 | MOHURD | Green Construction Technology Guidelines (Trial) | Guidelines |
| P21 | 2021/7/28 | MOHURD | Notice on Issuing a List of Replicable Practices for the Coordinated Development of Intelligent Construction and New Building Industrialization (First Batch) | Notice |
| P22 | 2021/9/10 | MIIT;<br>Office of the Central Cyberspace Affairs Commission (CAC);<br>MST;<br>MEE;<br>MOHURD;<br>Ministry of Agriculture and Rural Affairs (MoAR);<br>National Health Commission (NHC);<br>National Energy Bureau (NEB) | Three-Year Action Plan for the Construction of New Infrastructure for the Internet of Things (2021–2023) | Plan |
| P23 | 2021/10/10 | SC | National Outline for Standardization Development | Outline |
| P24 | 2021/10/21 | SC | Opinions on Promoting Green Development in Urban and Rural Construction | Opinion |
| P25 | 2021/10/26 | SC | Notice on the Issuance of the Action Plan for Carbon Peaking Before 2030 | Notice |
| P26 | 2021/11/22 | MOHURD | Notice on Publishing Typical Cases of New Technologies and Products for Intelligent Construction (First Batch) | Notice |
| P27 | 2022/1/19 | MOHURD | "14th Five-Year" Development Plan for the Construction Industry | Plan |
| P28 | 2022/10/1 | MOF;<br>MOHURD;<br>MIIT | Notice on Expanding the Scope of Government Procurement Policies to Support Green Building Materials and Improve Building Quality | Notice |
| P29 | 2022/10/12 | MOHURD | Notice on Announcing Intelligent Construction Pilot Cities | Notice |
| P30 | 2022/11/18 | MST;<br>MOHURD | "14th Five-Year" Special Plan for Urbanization and Urban Development Technological Innovation | Plan |

as a critical dimension, analyzing the implementation scope and effectiveness through the examination of target agencies and their operational levels.

Finally, the "Policy Disclosure" assesses the public accessibility of policies and their potential for broad societal impact, completing the comprehensive evaluation framework.

This integrated approach enables a systematic analysis of CICP, ensuring robust evaluation of their formulation quality, implementation mechanisms, and potential impacts across various sectors and governance levels while maintaining the flexibility to adapt to the evolving nature of intelligent construction development.

**Table 4. Example of encoding rules for text clauses.**

| ID | Policy Text | Contents of Clauses | Encoding |
|---|---|---|---|
| P15 | Guidance on Promoting the Coordinated Development of Intelligent Construction and Construction Industrialization | 3. Development goals<br>China aims to enhance intelligent construction by 2025, focusing on digitalization, intelligent platforms, technological advancement, and labor productivity while reducing resources and emissions. These efforts will transform SMEs and establish "Chinese Construction" as a global leader by 2035. | 15−3 |
| P27 | "14th Five-Year" Development Plan for the Construction Industry | 2. Development goals<br>(2) Development goals during the "14th Five-Year" period<br>In the "14th Five-Year" period, the plan targets over 30% prefabricated buildings in new structures, aiming to develop intelligent construction and building industrialization, including establishing industry platforms, creating construction robots, and cultivating industry bases. | 27-2-2 |
| P29 | Notice on Announcing Intelligent Construction Pilot Cities | Beijing and 24 other cities have been designated intelligent construction pilot cities, with a three-year pilot period starting from the announcement date. For details, see the attached list. | 29−0 |

**Table 5. Evaluation Index System and Criteria for CICP.**

| Primary Variable | Secondary Variable | | | | | | Evaluation Criteria |
|---|---|---|---|---|---|---|---|
| X1 Policy Scope | X1:1 Prediction | X1:2 Regulation | X1:3 Suggestion | X1:4 Support | X1:5 Guidance | X1:6 Other | Score 1 for fitting each secondary variable description; otherwise, score 0. (Note: In issuing institution, add a secondary variable for statistical calculation in case of joint documents issued by multiple departments.) |
| X2 Policy Effectiveness | X2:1 Short-term | X2:2 Medium-term | X2:3 Long-term | | | | |
| X3 Policy Field | X3:1 Economy | X3:2 Society | X3:3 Technology | X3:4 Environment | X3:5 Other | | |
| X4 Policy Perspective | X4:1 Macro | X4:2 Meso | X4:3 Micro | | | | |
| X5 Policy Function | X5:1 Government Procurement | X5:2 Standard Guidance | X5:3 System Constraint | X5:4 Technological Innovation | X5:5 Industry Cultivation | | |
| X6 Incentives and Constraints | X6:1 Investment Subsidies | X6:2 Legal Protection | X6:3 Talent Incentives | X6:4 Tax Benefits | X6:5 Intellectual Property | X6:6 Service Support | |
| X7 Policy Evaluation | X7:1 Based on Adequate | X7:2 Clear Objectives | X7:3 Detailed Planning | X7:4 Scientific Scheme | | | |
| X8 Issuing Institution | X8:1 SC | X8:2 Central Ministries | | | | | |
| X9 Policy Recipients | X9:1 Ministries of SC | X9:2 Provinces | X9:3 Affiliated Institutions | X9:4 Others | | | |
| X10 Policy Disclosure | | | | | | | Score 1 for public disclosure, 0 otherwise. |

**Building the multi-input–output tables and setting parameter values.** This research involves calculations with previously defined primary variables, constructing multi-input-output tables for the thirty policies (see Table 6), and integrating these with the standard PMC index formula. The values for secondary variables (Table 5), are assigned based on the policy texts' content analysis, following the methodology outlined in Formulas (1) to (2). All secondary variables

**Table 6. Multi-input–output table.**

| X1 | | | | | | X2 | | | X3 | | | | |
|---|---|---|---|---|---|---|---|---|---|---|---|---|---|
| X1:1 | X1:2 | X1:3 | X1:4 | X1:5 | X1:6 | X2:1 | X2:2 | X2:3 | X3:1 | X3:2 | X3:3 | X3:4 | X3:5 |
| X4 | | | X5 | | | | | X6 | | | | | |
| X4:1 | X4:2 | X4:3 | X5:1 | X5:2 | X5:3 | X5:4 | X5:5 | X6:1 | X6:2 | X6:3 | X6:4 | X6:5 | X6:6 |
| X7 | | | | X8 | | X9 | | | | X10 | | | |
| X7:1 | X7:2 | X7:3 | X7:4 | X8:1 | X8:2 | X9:1 | X9:2 | X9:3 | X9:4 | | | | |

are considered equally important and are thus weighted the same. When a policy encompasses a particular secondary variable, its parameter value is set to 1; otherwise, it is set to 0.

   **Calculating and detailed description of the PMC index.** Each primary variable is calculated using Formula (3), and subsequently, the PMC index is determined through Formula (4) based on these values. The 30 policy texts are then rated according to the PMC index, as the rating criteria outlined in the methodology section. Then, the PMC indices and corresponding ratings for the 30 policy texts are derived, as indicated in Table 7. The findings indicate:

(1) The average PMC index for 30 policies stands at 6.93, classified within the "Acceptable" at the policy evaluation level, closely approaching the "Positive" benchmark. The result suggests that CICP are well-rounded, addressing various dimensions and establishing a solid foundation for effective implementation.

(2) Each of the 30 policies has a PMC index exceeding 5.00, indicating the absence of poorly formulated policies.

(3) Within the quality evaluations of these 30 policies, two achieved "Excellent" (7%), fourteen "Positive" (46%), eight "Acceptable" (27%), and six "Average" (20%). These outcomes demonstrate that the formulation quality of 80% of the policies is considered acceptable, with over 50% having a constructive impact, reflecting well on the general formulate quality of CICP.

(4) Among these policies, "Decision on Accelerating the Cultivation and Development of Strategic Emerging Industries"(P1) [67] introduced by the State Council in 2010 marks the inception of the CICP framework, bearing a PMC index of 7.12 and classified as "Positive." The policy signifies its beneficial guidance on China's intelligent construction evolution.

(5) The highest quality of policy formulation was observed in the "Guiding Opinions on Promoting the Coordinated Development of Intelligent Construction and Construction Industrialization" (P15) [68], issued in 2020 by the MHURD along with 13 other departments, achieving a PMC index of 8.36 and classified as "Excellent." The policy indicates its significant role in fostering the rapid advancement of intelligent construction in China.

   **Classification and definition of policy phases.** Statistics based on the data in Table 7, a comparative analysis of policy evaluations across different periods reveals that the introduction of policy "P15" in 2020 marks a pivotal milestone in the evolution of CICP. As illustrated in Fig 4, the analysis further reveals a significant shift in the development of CICP occurred around the year 2020.

   The pre-"P15" period (2010–2019) was characterized by a gradual development pace, during which the government issued 14 intelligent construction-related policies, averaging 1.4 per year. The policies formulated in this phase had PMC index values ranging from 5.57 to 7.65, with 50% classified as "Acceptable" and 36% as "Positive," but none achieving "Excellent" ratings.

   In contrast, the post-"P15" period (2020–2022) witnessed a substantial acceleration in policy development, with the government releasing 16 policies within three years, averaging 5.3 policies annually. These policies demonstrated improved quality, with PMC index values ranging from 5.24 to 8.36, including 6% rated as "Acceptable," 56% as "Positive," and 13% as "Excellent."

**Table 7. Calculation and evaluation results of PMC index for 30 policy texts.**

|  |  | X1 | X2 | X3 | X4 | X5 | X6 | X7 | X8 | X9 | X10 | PMC Index | Concavity Index | Evaluation Class | Ranking |
|---|---|---|---|---|---|---|---|---|---|---|---|---|---|---|---|
| CICP (2010-2022) | P1 | 0.50 | 0.67 | 0.40 | 0.67 | 0.80 | 0.83 | 1.00 | 0.50 | 0.75 | 1.00 | 7.12 | 2.88 | Positive | 15 |
|  | P2 | 0.50 | 0.67 | 0.40 | 1.00 | 0.67 | 0.00 | 1.00 | 0.67 | 0.75 | 1.00 | 6.66 | 3.34 | Acceptable | 22 |
|  | P3 | 0.50 | 0.67 | 0.60 | 0.67 | 0.60 | 0.67 | 0.75 | 0.50 | 0.75 | 1.00 | 6.71 | 3.29 | Acceptable | 21 |
|  | P4 | 0.67 | 0.67 | 1.00 | 0.33 | 0.20 | 0.50 | 1.00 | 0.50 | 1.00 | 1.00 | 6.87 | 3.13 | Acceptable | 18 |
|  | P5 | 0.50 | 1.00 | 0.80 | 0.67 | 0.40 | 0.17 | 0.60 | 0.67 | 1.00 | 1.00 | 6.81 | 3.19 | Acceptable | 19 |
|  | P6 | 0.67 | 0.67 | 0.80 | 0.67 | 0.60 | 0.67 | 1.00 | 0.50 | 1.00 | 1.00 | 7.58 | 2.42 | Positive | 8 |
|  | P7 | 0.50 | 0.67 | 0.20 | 0.67 | 0.80 | 0.50 | 1.00 | 0.50 | 0.75 | 1.00 | 6.59 | 3.41 | Acceptable | 23 |
|  | P8 | 0.50 | 0.67 | 0.60 | 0.67 | 0.80 | 0.50 | 0.75 | 0.50 | 0.75 | 1.00 | 6.74 | 3.26 | Acceptable | 20 |
|  | P9 | 0.50 | 1.00 | 0.40 | 0.67 | 0.80 | 0.83 | 0.75 | 0.50 | 1.00 | 1.00 | 7.45 | 2.55 | Positive | 11 |
|  | P10 | 0.67 | 1.00 | 0.20 | 0.33 | 0.20 | 0.33 | 1.00 | 0.50 | 0.50 | 1.00 | 5.73 | 4.27 | Average | 26 |
|  | P11 | 0.50 | 1.00 | 0.60 | 0.67 | 0.60 | 0.50 | 0.75 | 0.50 | 1.00 | 1.00 | 7.12 | 2.88 | Positive | 15 |
|  | P12 | 0.50 | 0.67 | 0.60 | 1.00 | 0.80 | 0.83 | 1.00 | 0.50 | 0.75 | 1.00 | 7.65 | 2.35 | Positive | 6 |
|  | P13 | 0.50 | 1.00 | 0.60 | 0.33 | 0.80 | 0.67 | 0.75 | 0.50 | 0.75 | 1.00 | 6.90 | 3.10 | Acceptable | 17 |
|  | P14 | 0.67 | 1.00 | 0.20 | 0.33 | 0.20 | 0.17 | 1.00 | 0.50 | 0.50 | 1.00 | 5.57 | 4.43 | Average | 27 |
|  | P15 | 0.67 | 1.00 | 0.80 | 0.67 | 0.80 | 1.00 | 1.00 | 0.67 | 0.75 | 1.00 | 8.36 | 1.64 | Excellent | 1 |
|  | P16 | 0.50 | 0.67 | 0.40 | 0.67 | 1.00 | 0.50 | 1.00 | 0.67 | 0.75 | 1.00 | 7.16 | 2.84 | Positive | 13 |
|  | P17 | 0.50 | 1.00 | 0.60 | 0.67 | 0.80 | 0.83 | 1.00 | 0.67 | 0.75 | 1.00 | 7.82 | 2.18 | Positive | 4 |
|  | P18 | 0.50 | 0.67 | 0.40 | 0.67 | 0.80 | 0.67 | 1.00 | 0.67 | 1.00 | 1.00 | 7.38 | 2.62 | Positive | 12 |
|  | P19 | 0.50 | 0.33 | 0.40 | 0.33 | 0.60 | 0.40 | 0.75 | 0.50 | 0.50 | 1.00 | 5.31 | 4.69 | Average | 28 |
|  | P20 | 0.67 | 0.67 | 0.40 | 0.67 | 0.40 | 0.50 | 1.00 | 0.50 | 0.75 | 1.00 | 6.56 | 3.44 | Acceptable | 24 |
|  | P21 | 0.50 | 0.33 | 0.40 | 0.33 | 0.40 | 0.40 | 0.67 | 0.75 | 0.50 | 1.00 | 5.28 | 4.72 | Average | 29 |
|  | P22 | 0.67 | 0.67 | 0.60 | 0.67 | 0.80 | 0.83 | 1.00 | 0.67 | 0.75 | 1.00 | 7.66 | 2.34 | Positive | 5 |
|  | P23 | 0.67 | 1.00 | 0.40 | 0.67 | 0.60 | 0.67 | 1.00 | 0.50 | 1.00 | 1.00 | 7.51 | 2.49 | Positive | 10 |
|  | P24 | 0.50 | 1.00 | 0.80 | 0.67 | 0.60 | 0.50 | 1.00 | 0.50 | 1.00 | 1.00 | 7.57 | 2.43 | Positive | 9 |
|  | P25 | 0.50 | 1.00 | 0.80 | 0.67 | 0.80 | 1.00 | 1.00 | 0.50 | 1.00 | 1.00 | 8.27 | 1.73 | Excellent | 2 |
|  | P26 | 0.50 | 0.33 | 0.40 | 0.33 | 0.60 | 0.33 | 0.75 | 0.50 | 0.50 | 1.00 | 5.24 | 4.76 | Average | 30 |
|  | P27 | 0.67 | 0.67 | 0.80 | 1.00 | 0.80 | 0.67 | 1.00 | 0.50 | 0.75 | 1.00 | 7.86 | 2.14 | Positive | 3 |
|  | P28 | 0.50 | 0.67 | 0.60 | 0.67 | 0.60 | 0.67 | 1.00 | 0.67 | 0.75 | 1.00 | 7.13 | 2.87 | Positive | 14 |
|  | P29 | 0.50 | 0.33 | 0.60 | 0.67 | 0.60 | 0.33 | 0.75 | 0.50 | 0.50 | 1.00 | 5.78 | 4.22 | Average | 25 |
|  | P30 | 0.50 | 0.67 | 0.60 | 1.00 | 0.80 | 0.60 | 1.00 | 0.67 | 0.75 | 1.00 | 7.59 | 2.41 | Positive | 7 |
| **Average in Overall** |  | 0.55 | 0.75 | 0.55 | 0.63 | 0.64 | 0.57 | 0.91 | 0.56 | 0.78 | 1.00 | 6.93 | 3.07 | Acceptable |  |

This comparative analysis indicates a transformative paradigm shift in CICP development following the introduction of Policy "P15", manifesting in both quantitative and qualitative dimensions. The substantial increase in policy issuance frequency (from 1.4 to 5.3 policies per year) and the emergence of "Excellent" rated policies (from 0% to 13%) collectively demonstrate enhanced policy-making capacity and strategic focus.

The empirical analysis above substantiates that CICP demonstrates distinct phased development characteristics with identifiable transition points. Building upon data-driven evidence, our comprehensive content analysis of 30 CICP texts further validates this developmental trajectory, revealing a systematic progression from the Cultivation and Exploration Phase (CEP: 2010–2019) to the Development and Promotion Phase (DPP: 2020–2022), as delineated in Fig 5.

During the foundational CEP period, the Chinese government established critical policy frameworks through landmark documents such as the "Decision on Accelerating the Cultivation and Development of Strategic Emerging Industries" (P1), "Outline for the Development of Information Technology in the Construction Industry (2011-2015)" (P3), "Made in China

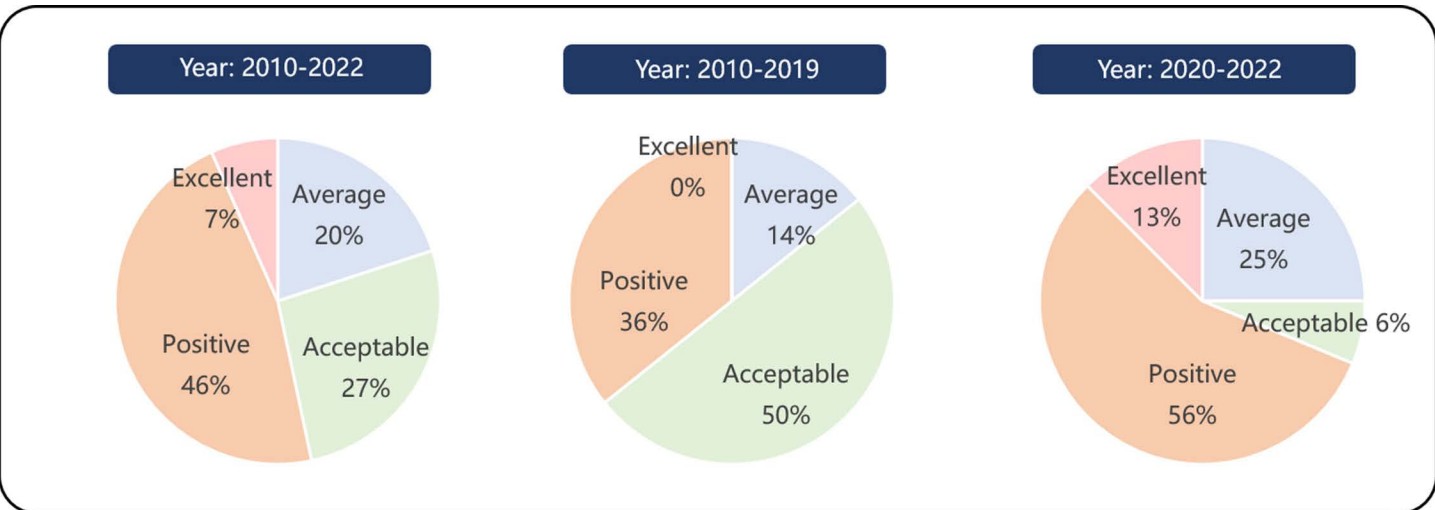

**Fig 4. Comparison of policy ratings in different periods.**

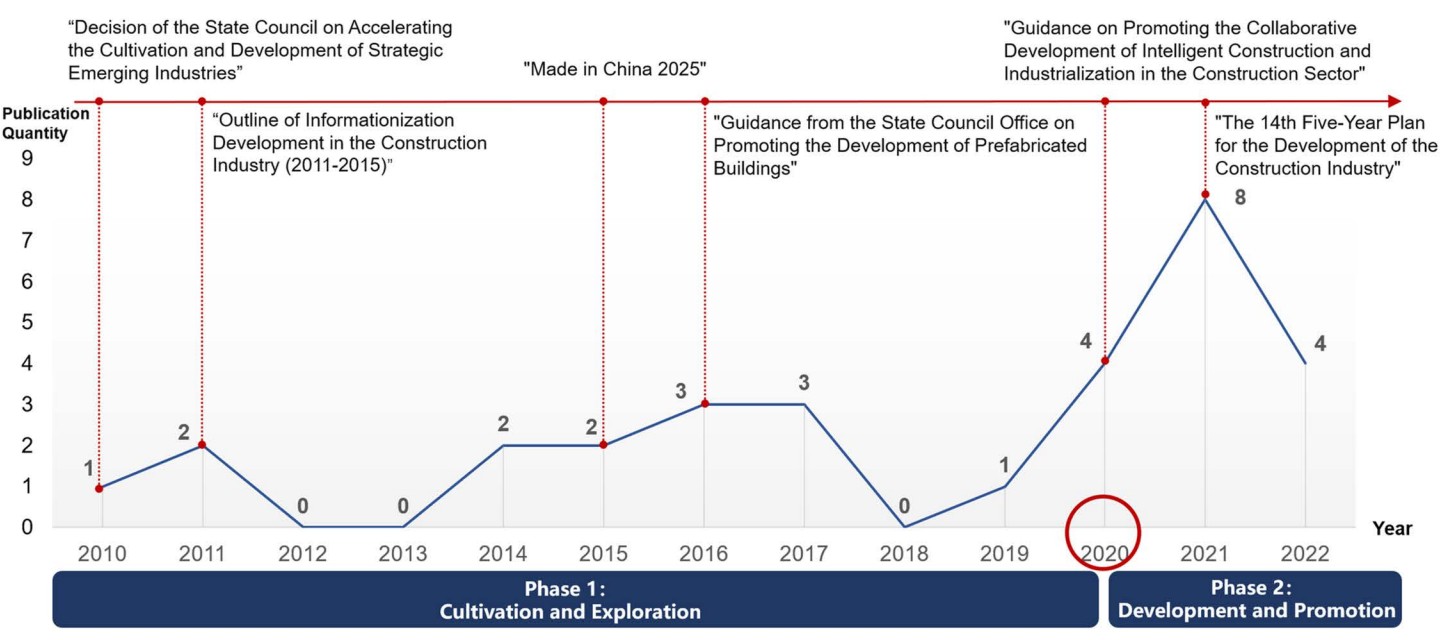

**Fig 5. Policy phases.**

2025" (P6), and "Guidance on Vigorously Developing Prefabricated Buildings"(P9). These initiatives primarily focused on technological infrastructure development, particularly promoting digital technologies including BIM, 3D printing, and IoT in prefabricated construction. Through national-level innovation support and industrial guidance, CEP facilitated the construction industry's initial transition toward informatization and digital transformation. However, the phase was characterized by restricted policy scope, practical application challenges, and noticeable stagnation periods during 2012-2013 and 2018-2019, reflecting its exploratory nature and limited implementation breadth.

The subsequent DPP represents a transformative phase initiated by the 2020 policy "Guiding Opinions on Promoting the Coordinated Development of Intelligent Construction and Construction Industrialization" (P15), which introduced comprehensive strategies emphasizing construction industrialization, digital transformation, and intelligent system integration. This phase reached a significant milestone with the 2022 "14th Five-Year Plan for Construction Industry Development" (P25) [69], establishing ambitious objectives for global leadership in intelligent construction. Characterized by extended temporal efficacy, expanded application domains, clearer strategic objectives, diversified implementation tools, and enhanced synergy with construction industrialization, DPP policies have significantly accelerated intelligent construction adoption across China's building sector.

This evolution from CEP to DPP reflects a fundamental shift from foundational policy frameworks to comprehensive implementation strategies, demonstrating CICP's adaptive capacity in responding to technological advancements and industry demands. The phase transition marks a critical development in the CICP landscape, moving from exploratory experimentation to rapid development and maturation, while establishing a robust foundation for future policy innovation and industry transformation.

**Discussion on phase shift.**  Table 8 presents the comparative PMC index analysis across two phases. The CEP yielded an average score of 6.82 ("Acceptable"), while DPP achieved 7.03 ("Positive"), demonstrating enhanced policy quality. Fig 6 reveals significant rating distribution shifts: "Average" policies increased from 2 to 4, "Acceptable" decreased from 7 to 1, "Positive" rose from 5 to 9, and "Excellent" emerged from 0 to 2. These improvements indicate DPP's stronger impact in advancing China's intelligent construction. At the same time, China's intelligent construction market has also risen rapidly from 150 billion yuan in 2019–250 billion yuan in 2022, with an average annual growth rate of more than 20%. This is further evidence of the effectiveness of the DPP policy.

The comprehensive content analysis of CICP texts also reveals a significant evolution in policy formulation approaches, transitioning from the exploratory and tentative strategies of the initial phase (CEP) to more systematic, comprehensive, and sustainable frameworks in DPP, with stronger integration of practical applications. This transformative shift holds profound implications for CICP development, as evidenced by several critical dimensions of policy improvement.

**Table 8.  The average PMC index values of the policies in two phases.**

|  | PMC Index | Concavity Index | Evaluation Class | Ranking |
|---|---|---|---|---|
| Average in CEP | 6.82 | 3.18 | Acceptable |  |
| Average in DPP | 7.03 | 2.97 | Positive |  |

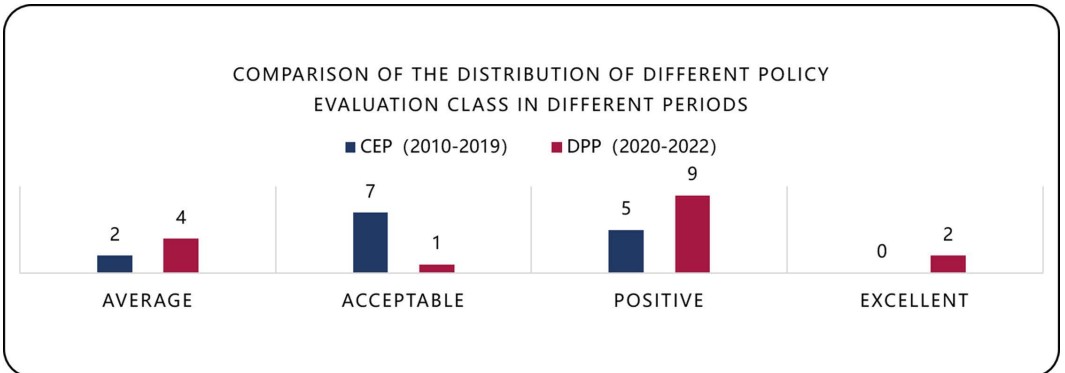

**Fig 6.  Comparison of policy ratings between CEP and DPP.**

Firstly, the transition to a more practical policy approach represents a fundamental paradigm shift from experimental technology promotion to mainstream application of intelligent construction principles. The introduction of "P15", with its emphasis on construction industrialization and digitalization, has profoundly impacted construction practices by encouraging widespread adoption of new technologies and implementation of standardized processes across the industry. This shift is particularly evident in the increased utilization of BIM technologies and prefabricated construction methods, which have become more prevalent since the implementation of DPP policies.

Secondly, the DPP's focus on long-term goals and sustainable growth, exemplified by Policy "P25", directly addresses the limitations observed in CEP, where policy stagnation and fragmented approaches had previously slowed progress. By establishing a comprehensive, forward-looking agenda, the DPP ensures greater scalability and sustainability of CICP. This is achieved through the introduction of financial and tax incentives, along with targeted support for Small and Medium Enterprises (SMEs), which effectively address the barriers that hindered the broader adoption of intelligent construction technologies during the first phase.

In addition, the shift from CEP to DPP was driven by several key factors, including the maturation of core intelligent construction technologies, accumulated implementation experience from pilot projects, and the construction industry's increasing capacity for digital transformation. These developments enabled the formulation of more effective policies that better align with industry needs and technological capabilities. Especially, the DPP policies demonstrate enhanced effectiveness through their holistic nature, integrating economic, environmental, and social objectives while maintaining clear implementation guidelines and support mechanisms. This comprehensive approach has resulted in policies that are not only more frequent and higher in quality but also more responsive to the evolving needs of China's construction industry.

However, the absence of any policy receiving a "perfect" rating indicates room for further optimization in CICP. Before offering targeted recommendations for enhancement, in-depth reviews and discussions of the evolution, status, and trends of the CICP are imperative. These discussions will integrate the historical context of policy evolution and its characteristic patterns, critically analyzing the factors contributing to DPP's superior effectiveness compared to CEP, while identifying deficiencies and potential improvement areas within current policies. The analysis in detail will employ a dynamic analytical framework to undertake a comparative examination of policy features within the CEP and DPP across various dimensions, including types of policies, entities responsible for policy formulation, and the goals and tools of these policies. These analytical dimensions maintain conceptual consistency with the evaluation variables established in the PMC model: text type analysis corresponds to "Policy Scope", intergovernmental relationship examination aligns with "Issuing Institution" and "Policy Recipients", content theme analysis relates to "Policy Fields", and objective-tool analysis connects with "Policy Function" along with "Incentives and Constraints", and so on. By leveraging the analytical outcomes derived from the PMC model, this methodical examination will culminate in scientifically grounded recommendations for policy optimization.

## Discussions of policy evolution and status

**Types of policy text.** Policy texts vary in effectiveness, with "notifications" and "opinions" offering broad applicability and timeliness, significantly influencing recipients' actions. Conversely, "guidelines," "plans," and "manuals" are more operational, guiding practical work. This research statistically analyzes the distribution of 30 policy texts (Table 9), comparing policies' binding and operational nature across different phases (Fig 7). The results show a predominance of "opinions," "notifications," "plans," and "outlines," indicating the government's promotional solid intent. However, the operational aspect of policies requires further improvement. From the CEP to DPP, the share of policies with strong constraints increased from 77% to 88%, while those with solid operational characteristics rose from 38% to 62%. This shift is primarily due to "notifications," which act as action guides, replacing "opinions," known for suggestion-based guidance. This transition suggests a more precise government policy direction, focusing on optimizing institutional measures and detailed tasks to enhance policy effectiveness.

**Table 9. Distribution of text types.**

| Text Type | Quantity (copies) | Proportion(%) | Constraints | Operability |
|---|---|---|---|---|
| Opinion | 9 | 30.00 | Strong | Weak |
| Notification | 6 | 20.00 | Strong | Strong |
| Planning | 5 | 16.67 | Strong | Weak |
| Outline | 3 | 10.00 | Weak | Strong |
| Standard | 2 | 6.68 | Strong | Strong |
| Guideline | 1 | 3.33 | Strong | Strong |
| Handbook | 1 | 3.33 | Weak | Strong |
| Scheme | 1 | 3.33 | Weak | Strong |
| Program | 1 | 3.33 | Weak | Strong |
| Decision | 1 | 3.33 | Strong | Weak |

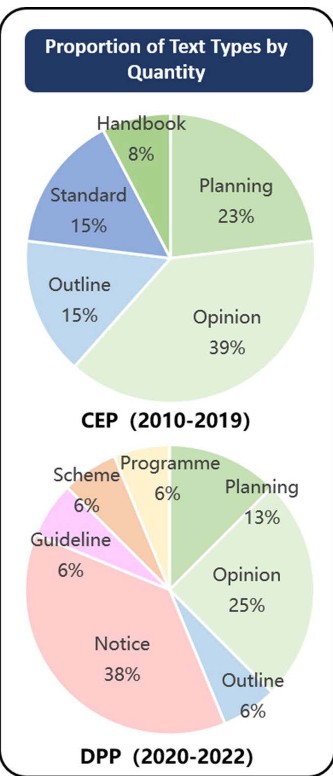
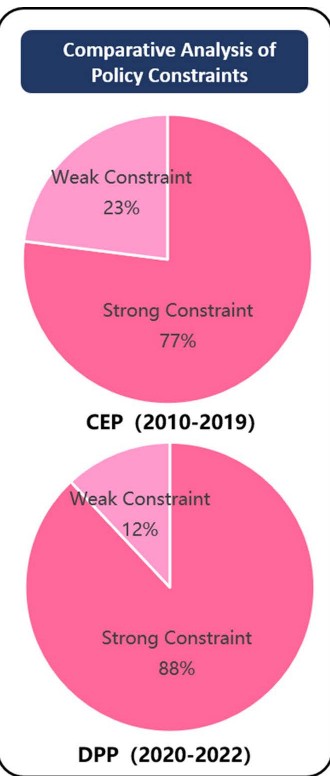
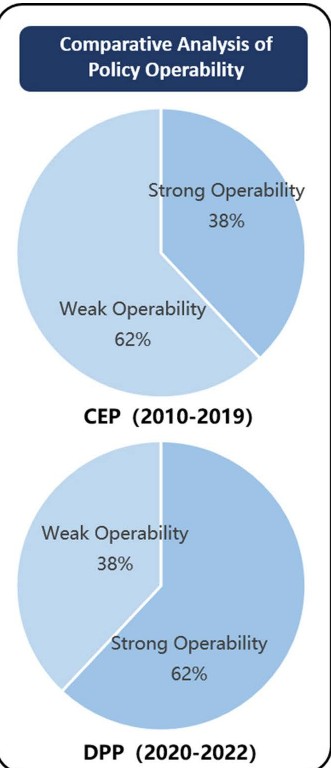

**Fig 7. Statics of text types.**

**Department issuing policy.** Policy formulation involves interactions among government departments as critical actors. Analyzing their co-authorship network in issuing policies uncovers patterns of cooperation and conflict [15]. The results show (Fig 8):

(1) Inter-departmental collaboration in policy formulation is intensifying. Of the policies analyzed, 70% were issued by individual departments, notably the SC and the MOHURD. Meanwhile, 30% involved multi-departmental cooperation, especially during the DPP. This trend highlights the need for varied expertise in integrating intelligent construction with industrialization.

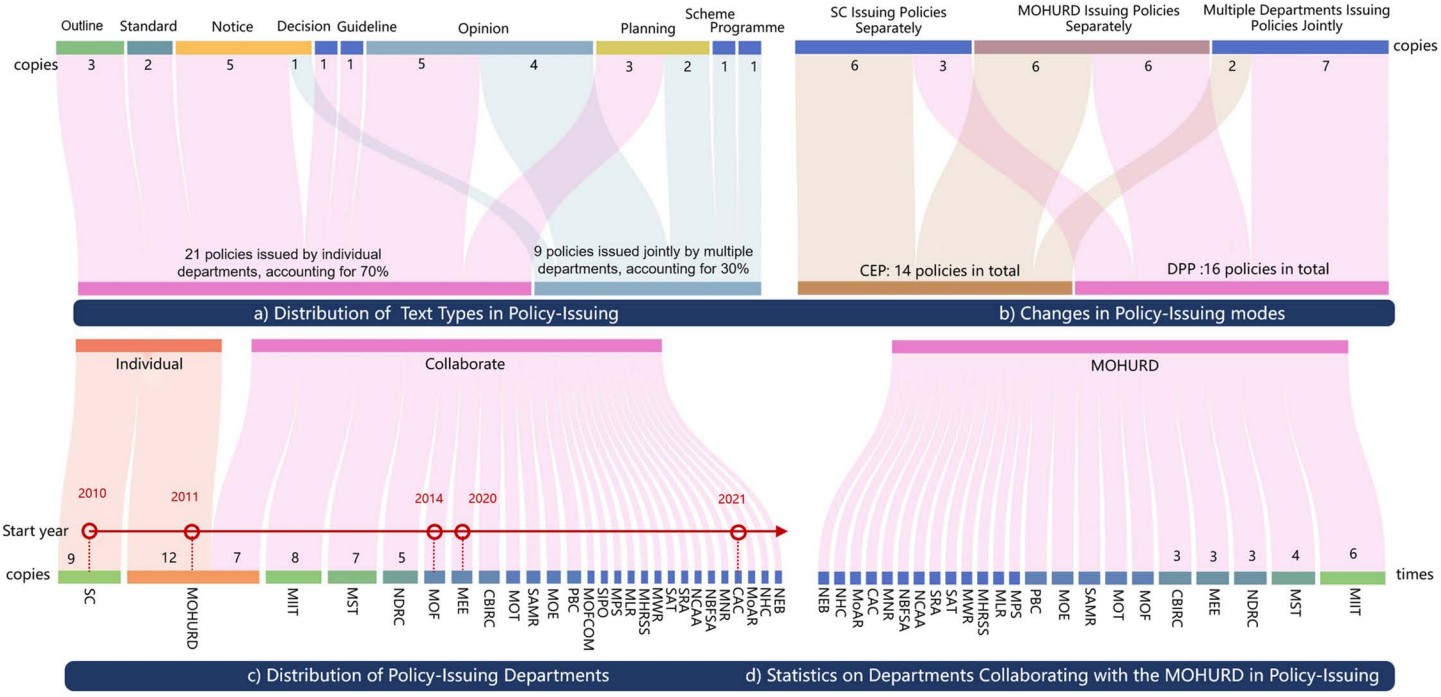

**Fig 8. Analysis of intergovernmental relations.**

(2)  Different phases show varied departmental involvement. During CEP, the SC led the initiative, focusing on technological innovation with contributions from MOHURD, the Ministry of Industry and Information Technology (MIIT), the Ministry of Science and Technology (MST), and the National Development and Reform Commission (NDRC). The participation was broader in DPP.

(3)  Different departments have varying degrees of collaboration. MOHURD, as the central leading department, collaborates closely with MIIT, NDRC, the Ministry of Ecology and Environment (MEE), and the China Banking and Insurance Regulatory Commission (CBIRC). This phenomenon demonstrates that industrial, technological, ecological, and economic support are significant principles in government policy formulation.

**Themes of policy**

(1)  High-frequency words statistics

Analyzing high-frequency thematic words in policy texts helps focus on core themes. Word distribution in the title indicates policy formulation's macro-direction, while content word distribution reflects key implementation points. Using ROSTCM6 software, 30 policy texts from Table 1 underwent word segmentation. High-frequency words were extracted, showing frequency distribution (see Tables 10 and 11). The word frequency of titles suggests those policies primarily aimed at advancing intelligent construction in the industry. During the CEP, the focus was on informatization, shifting to green and intelligent construction in the DPP. The word frequency of content highlights technological innovation to promote intelligent construction. Words like "standard, " "management, " "service, " "system, " "mechanism, " "model, " "platform, " and "data" are prominent, aligning with Chinese experts' and scholars' policy recommendations [3,9–11].

**Table 10. High-frequency words statistics in titles.**

| Overall (2010–2022) | | | | CEP (2010–2019) | | DPP (2020–2022) | |
|---|---|---|---|---|---|---|---|
| High-Frequency Words (Top 20) | | | | High-Frequency Words (Top 10) | | High-Frequency Words (Top 10) | |
| Theme Word | Frequency | Theme Word | Frequency | Theme Word | Frequency | Theme Word | Frequency |
| Development | 15 | Green | 4 | Development | 8 | Development | 7 |
| Construction | 9 | Planning | 4 | Opinion | 5 | Construction | 6 |
| Opinion | 9 | Urban and Rural | 4 | Construction | 4 | Intelligent | 5 |
| Building | 6 | Outline | 3 | Construction Industry | 4 | Notice | 5 |
| Intelligent | 6 | Action | 3 | Guidance | 3 | Urban and Rural | 5 |
| Guidance | 5 | Promote | 3 | Outline | 2 | Green | 4 |
| Notice | 5 | Industrialization | 3 | Information Model | 2 | MOHURD | 4 |
| Construction Industry | 5 | SC | 3 | Promote | 2 | Action | 4 |
| New Type | 4 | MOHURD | 3 | Informatization | 2 | Opinion | 3 |
| SC | 4 | Strategic | 2 | Development | 2 | Development | 3 |

**Table 11. High-frequency words statistics in contents.**

| Overall (2010–2022) | | | | CEP (2010–2019) | | DPP (2020–2022) | |
|---|---|---|---|---|---|---|---|
| High-Frequency Words (Top 50) | | | | High-Frequency Words (Top 25) | | High-Frequency Words (Top25) | |
| Theme Word | Frequency | Theme Word | Frequency | Theme Word | Frequency | Theme Word | Frequency |
| Technology | 1449 | Foundation | 354 | Technology | 1001 | Architecture | 509 |
| Development | 1106 | Level | 349 | Development | 681 | Technology | 448 |
| Construction | 979 | Enhance | 345 | Enterprise | 596 | Green | 435 |
| Architecture | 961 | Project | 339 | Construction | 581 | Development | 425 |
| Engineering | 856 | Construction | 333 | Engineering | 564 | Construction | 398 |
| Enterprise | 767 | Improve | 329 | Management | 526 | Standard | 361 |
| Application | 723 | Mechanism | 327 | Application | 502 | Engineering | 292 |
| Management | 713 | Data | 319 | Architecture | 452 | Promote | 233 |
| Standard | 672 | Material | 315 | Model | 449 | Advance | 225 |
| Service | 562 | Environment | 308 | Service | 421 | Application | 221 |
| Green | 560 | Facility | 295 | System | 401 | Strengthen | 193 |
| Promote | 501 | Intelligent | 287 | Standard | 311 | Perfect | 190 |
| System | 491 | Realize | 285 | Innovation | 299 | Management | 187 |
| Model | 476 | Utilize | 281 | Establish | 276 | Accelerate | 178 |
| Innovation | 465 | Capability | 276 | Advance | 276 | Fabrication | 177 |
| Advance | 444 | Market | 258 | Resource | 275 | Quality | 174 |
| Establish | 437 | Key | 249 | Safety | 269 | Enhance | 174 |
| Safety | 424 | Research | 248 | Data | 260 | Intelligent | 172 |
| Strengthen | 423 | Platform | 244 | Material | 255 | Enterprise | 171 |
| Design | 413 | Carry out | 243 | Perfect | 254 | Design | 169 |
| Accelerate | 399 | New Type | 242 | Design | 244 | System | 167 |
| Quality | 397 | Institution | 240 | Strengthen | 230 | Innovation | 166 |
| Resource | 395 | Structure | 237 | Quality | 223 | Standardization | 164 |
| Promote | 393 | Field | 235 | Accelerate | 221 | Foundation | 163 |
| System | 385 | Planning | 235 | System | 218 | Construction | 160 |

(2) Semantic network analysis

This research explores the associative structures in CEP and DPP policy texts using ROSTCM6's semantic network analysis and Table 11's high-frequency word statistics. The resulting concentric semantic network diagram (Fig 9) uses nodes to represent high-frequency words, with their connections indicating co-occurrence frequency. The more links a node has to other nodes, the more significant it is. During CEP, the policies focused on constructing infrastructure, improving management systems, strengthening technological innovation, and standard setting. In DPP, policies are committed to accelerating the engineering application of intelligent technologies and promoting green and low-carbon construction.

**Policy goals and tools**

(1) Policy goals

This research classifies the goals of CICP into seven categories, concluding objectives and tasks at the technology, construction, industry, service, and application level. (Fig 10) Following Table 2's approach, this research coded 30 policies' content clauses and quantified the policy clause provisions per goal based on the content analysis. (Table 12) The research process reveals policy shifts focus across phases (Fig 11). Key findings show consistent emphasis on "Enhancing technological innovation capability" (G2), "Improving digitalization, informatization, and intelligence level" (G3), and "Revolutionizing service and supervisory systems" (G7). The policy evolution highlights an initial focus on digitalizing construction and technology innovation in the CEP, shifting in the DPP to prioritize" accelerating the industrialization of construction" (G1), with increased attention on "promoting green construction" (G4), " establishing industrial systems" (G5), and "expanding application scenarios " (G6). This shift towards a more comprehensive and specific policy approach is evident, particularly in industrial and environmental applications.

(2) Policy tools

The classification of policy tools influences the achievement of policy goals. Considering the interpretation of CICP and referencing research [13,14,60], this research categorizes the tools of CICP into three main types and fifteen sub-tools, following Rothwell and Zegveld's model [70] of Supply-side (SP), Demand-side (DP), and Environmental Policy (EP) Tools. (Fig 12) SP Tools enhances intelligent construction by providing financial support, IT, talent development, and regulatory services. DP Tools promotes it through collaborative projects, pilot demonstrations, and enterprise support. EP Tools indirectly assist via strategic planning, legal frameworks, and financial incentives, fostering a conducive environment for intelligent construction growth.

The research merged the coded content clauses of 30 policies with the previously discussed classification of policy tools for intelligent construction, compiling their distribution in various phases. (Table 13) Subsequently, the proportions of policy tools in CEP and DPP were calculated and analyzed. (Fig 13) Policy tools' distribution highlights SP and EP tools' dominance at 39% and 37%, with DP tools at 24%, indicating the need for more intelligent construction promotion efforts. Usage among the 15 sub-tools varies. SP tools focus on system optimization (T1-5) and research infrastructure (T1-2, T1-4), while funding and talent development (T1-1, T1-3) require support. DP tools emphasize industry applications (T2-3), with minimal use of market mechanisms (T2-2, T2-5). EP tools, led by T3-3, reflect China's role in setting standards, suggesting an increase in institutional and financial development policy support. The shift from CEP to DPP indicates a strategic government change. SP tools show increased research (T1-2) but reduced talent development (T1-3). DP tools maintain a market approach (T2-1, T2-3, T2-4), while EP tools' changes (T3-3, T3-5) focus on practical policy implementation for intelligent construction.

(3) Coupling analysis of policy goals and tools

A matrix model coupling three primary policy tools, fifteen sub-tools, and seven goals illustrates the evolution of government strategies across two phases. (Fig 14) Transitioning from CEP to DPP, policy integration with primary tools has increased significantly.

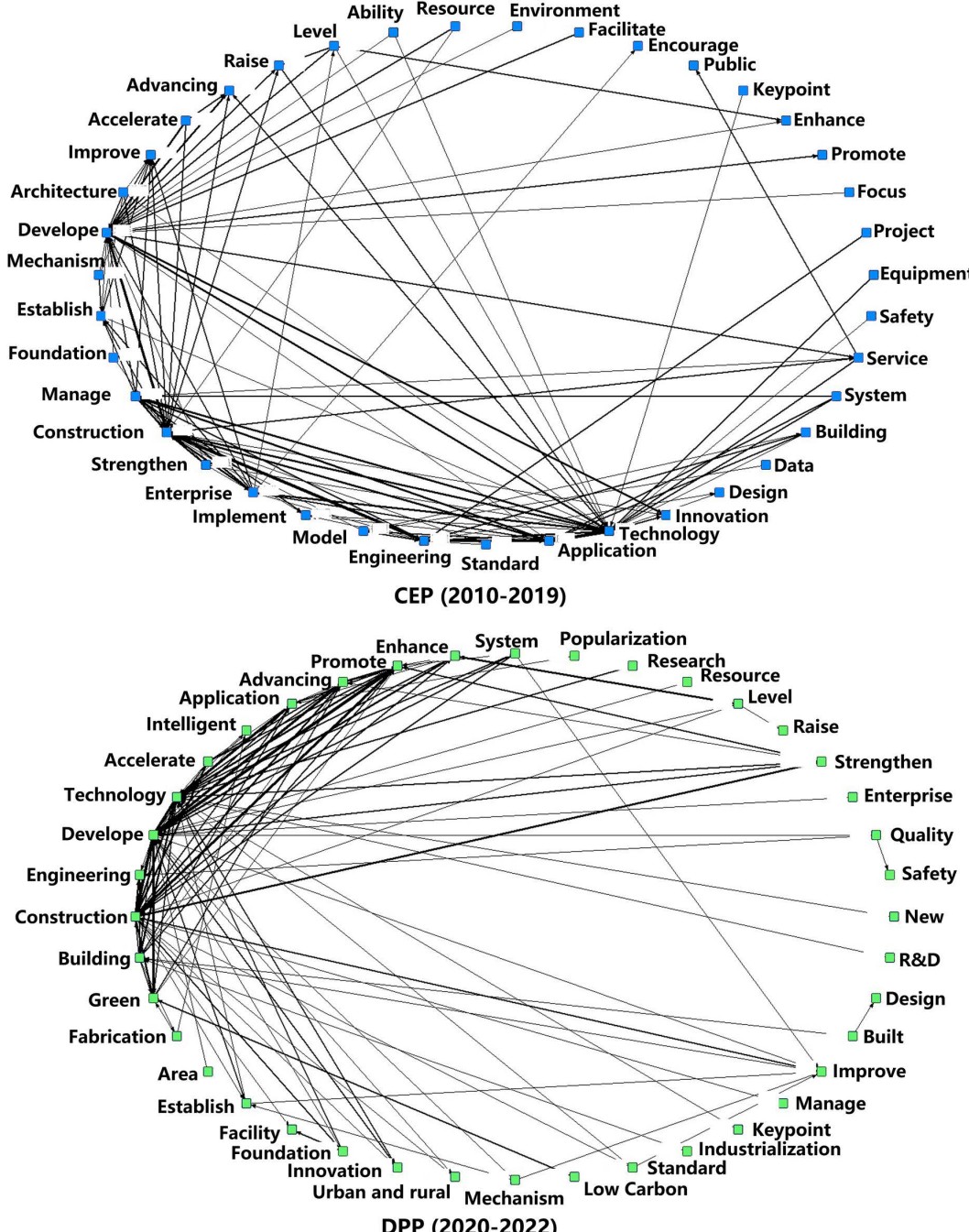

**Fig 9. Capacitive analysis of high-frequency words in text content between CEP and DPP.**

SP tools grew from 33.33% to 43.48%, DP tools from 20.37% to 31.62%, while EP tools fell from 46.30% to 24.90%, suggesting a shift towards comprehensive development. In CEP, EP tools led, focusing on resource optimization and institutional mechanisms. SP tools like T1-4 and T1-5 supported goals like G7, while significant DP tools, like T2-3, aided multiple goals. SP tools gained prominence in DPP, particularly T1-2, aiding G3 and G2, while DP tools like T2-3 and T2-4

| Accelerate the Upgrade of Construction Industrialization (G1) | Enhance Technological Innovation Capability (G2) | Improve Digitalization, Informatization, and Intelligentization Level (G3) |
|---|---|---|
| □ Develop prefabricated construction and establish a specialized, large-scale, and informationalized production system.<br>□ Enhance the integrated innovation and application of core technologies such as BIM, IoT, and AI.<br>□ Advance the R&D, as well as the application of intelligent construction equipment to improve labor productivity.<br>□ Establish the internet platform for the construction industry and promote the application of intelligent production lines for prefabricated components. | □ Promote the R&D of fundamental and key core technologies in intelligent construction and industrialization of construction.<br>□ Accelerate the R&A of construction robots, systemic software, data platforms, and integrated construction platforms.<br>□ Establish platforms for technological innovation.<br>□ Create standards system for intelligent construction and industrialization of construction. | □ Promote the adoption of digital collaborative design, establishing foundational digital design platforms and integrated systems.<br>□ Accelerate the integrated application of BIM technology to establish a technical framework and standards system.<br>□ Expedite the application of intelligent production, construction technology and equipment, and information management systems.<br>□ Foster the on-site application of construction robots and intelligent construction machinery. |

| Promote Green Construction Models (G4) | Establish Industrial System (G5) | Expand Application Scenarios (G6) | Revolutionize Service and Supervisory Systems (G7) |
|---|---|---|---|
| □ Deepen the pilot programs for green construction and initiate the creation of demonstration projects.<br>□ Expedite the breakthroughs in core technologies and their industrial applications, and establish guiding technical frameworks and comprehensive standards systems.<br>□ Develop a green supply chain in the construction industry, promote circular production methods, and enhance the comprehensive utilization of construction waste.<br>□ Intensify research and development in energy-saving and environmental protection technologies, processes, and equipment to improve energy efficiency levels. | □ Innovate the collaborative development of intelligent construction and industrialized building in terms of organizational methods, processes, and management models.<br>□ Cultivate general contracting enterprises to construct an open industrial system.<br>□ Establish industrial bases for intelligent construction and accelerate the building of a skilled workforce.<br>□ Create multi-party collaborative platforms for intelligent construction work. | □ Facilitate the transformation of scientific and technological achievements, integration and innovative application of major products, and demonstration applications.<br>□ Strengthen the exemplary leadership of projects and leading enterprises to expand the scope of technology application.<br>□ Publish catalogues of mature technologies to accelerate their widespread adoption.<br>□ Implement pilot demonstration actions for intelligent construction, summarize policies and models that can be replicated and promoted, and release cases of technological product innovation services. | □ Develop governmental service and decision-making information systems, establish mechanisms for big data-assisted scientific decision-making and market regulation.<br>□ Refine the digital delivery and review management systems for outcomes.<br>□ Create engineering quality and safety supervision models and mechanisms that align with intelligent construction.<br>□ Achieve interconnectivity between upstream and downstream enterprises in the supply chain to enhance collaborative efficiency. |

**Fig 10. Summary of policy goals.**

supported various goals. The most used EP tool was T3-3, helping goals like G4. This coupling analysis shows a strategic evolution in China's intelligent construction, initially emphasizing EP tools for a conducive environment, then transitioning to SP and DP tools for balanced, comprehensive development.

## Summary of discussions

The evolution of CICP reflects a dynamic interplay of strategic adaptation and institutional learning, marked by distinct phases of development. From the CEP to the DPP, the policy framework has transitioned from foundational experimentation to systematic implementation, driven by technological maturation, industry demands, and strategic national priorities.

Table 12. Statics of policy goal clauses in the two phases.

| ID of Goals | 0 (2010–2019) | | | | | | | | | | | | | | | | Number of Clauses | Proportion |
|---|---|---|---|---|---|---|---|---|---|---|---|---|---|---|---|---|---|---|
| | P1 | P2 | P3 | P4 | P5 | P6 | P7 | P8 | P9 | P10 | P11 | P12 | P13 | P14 | | | | |
| G1 | | | | 1 | 2 | 3 | | 1 | 1 | | | 1 | | | | | 9 | 8.57% |
| G2 | 2 | | 1 | | 3 | 4 | | 1 | 2 | | 1 | 3 | 1 | | | | 18 | 17.14% |
| G3 | 2 | 2 | 2 | 4 | 2 | 3 | 2 | 3 | 2 | 1 | 2 | 2 | | 1 | | | 28 | 26.67% |
| G4 | | | | 1 | | 3 | 1 | 1 | 2 | | | 3 | | | | | 11 | 10.48% |
| G5 | | | | | 1 | 1 | 1 | 1 | 2 | | 1 | 2 | 2 | | | | 11 | 10.48% |
| G6 | 2 | | | 1 | 1 | 2 | 1 | 4 | | | 1 | | | | | | 12 | 11.43% |
| G7 | 2 | | 1 | 2 | 1 | 1 | 1 | 1 | | 3 | 3 | 1 | | | | | 16 | 15.23% |
| ID of Goals | DPP (2020–2022) | | | | | | | | | | | | | | | | Number of Clauses | Proportion |
| | P15 | P16 | P17 | P18 | P19 | P20 | P21 | P22 | P23 | P24 | P25 | P26 | P27 | P28 | P29 | P30 | | |
| G1 | 5 | 1 | 6 | 1 | | | 1 | | | 1 | 1 | | 4 | 1 | 1 | 1 | 23 | 13.61% |
| G2 | 1 | 1 | 7 | 1 | 1 | | | 1 | 3 | 1 | 1 | 1 | 1 | | 1 | 4 | 24 | 14.20% |
| G3 | 5 | | 4 | 1 | 1 | 1 | 1 | 4 | 2 | 2 | | 1 | 1 | | | 4 | 28 | 16.57% |
| G4 | 1 | 2 | | 2 | | 1 | | | 1 | 4 | 1 | | 2 | 4 | | 3 | 21 | 12.43% |
| G5 | 6 | 1 | 8 | 1 | | | | 2 | 1 | 1 | 2 | | 2 | | | 3 | 27 | 15.98% |
| G6 | 6 | | 6 | 1 | 1 | | 1 | 3 | | 2 | 3 | | 1 | | | 2 | 26 | 15.38% |
| G7 | 1 | 4 | 3 | 2 | | | | | 1 | 1 | 3 | | 2 | 1 | | 2 | 20 | 11.83% |

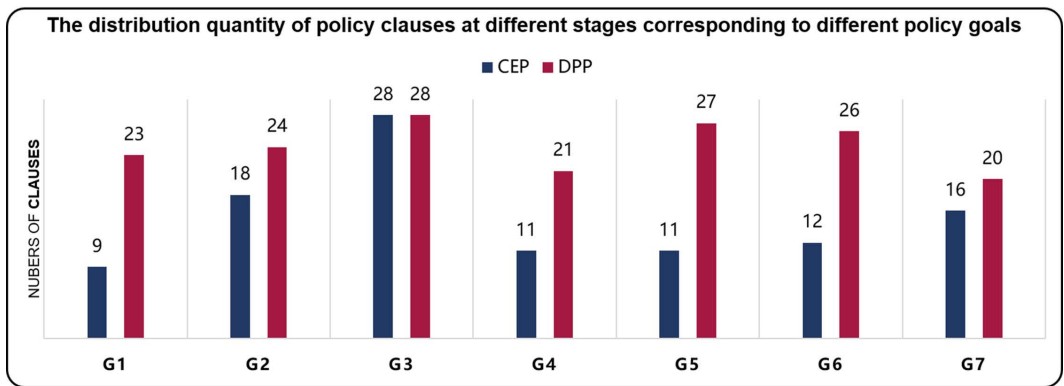

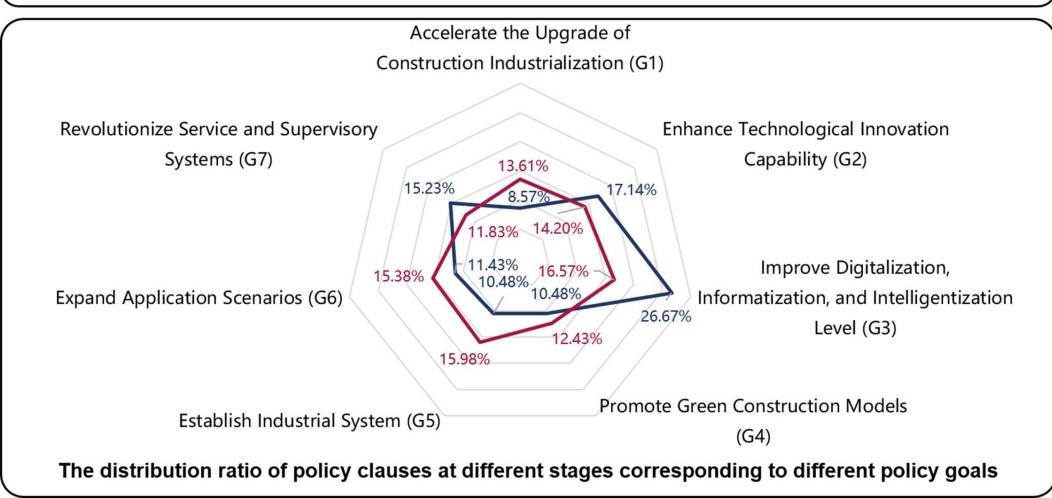

Fig 11. The proportional distribution of policy goals in the two phases.

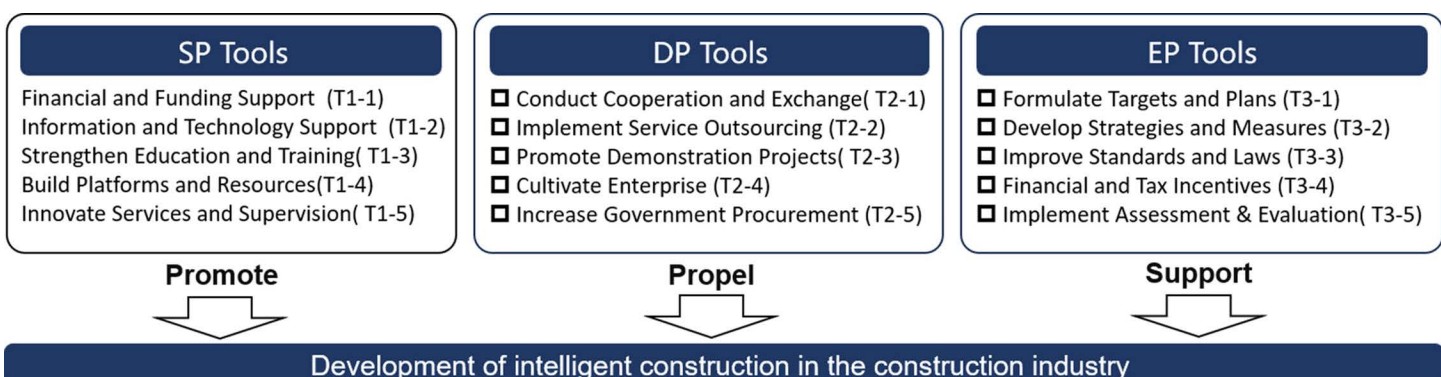

**Fig 12. Categorization of policy tools.**

(1) Features of CICP evolution

The CICP's evolution is characterized by a paradigm shift from fragmented, exploratory policies to integrated, goal-oriented strategies. During the CEP, policies focused on establishing digital infrastructure and promoting technologies like BIM and IoT, yet faced challenges in practical application and policy continuity. The DPP, initiated by Policy "P15", introduced a more cohesive approach, emphasizing industrialization, digitalization, and sustainability. This phase leveraged advanced tools such as tax incentives, SME support, and interdepartmental collaboration, resulting in accelerated technology adoption and industry transformation.

(2) Characteristics of CICP status

The current status of CICP reflects a more holistic and integrated approach to policy formulation. The DPP policies are characterized by increased inter-departmental collaboration, a broader range of policy tools, and a stronger emphasis on long-term goals such as green construction and industrial system development. The shift from EP tools to SP and DP tools in the DPP highlights a strategic move towards more direct and actionable policy measures, supported by financial incentives, research infrastructure, and industry applications. This rebalancing also aligns with broader objectives of economic growth, environmental sustainability, and technological leadership.

(3) Factors driving policy changes and DPP effectiveness

Firstly, technological advancements (e.g., BIM, IoT, and AI), accumulated implementation experience and heightened global competition. These factors addressed CEP limitations, such as policy stagnation and fragmented governance.

Secondly, enhanced interdepartmental collaboration, particularly between MOHURD, MIIT, and NDRC, enabled coordinated action on complex challenges.

Finally, financial mechanisms, including subsidies and tax incentives, directly tackled barriers to technology adoption, while policies like the "14th Five-Year Plan" (P25) institutionalized long-term objectives. The DPP's efficacy stems from its ability to align policy tools with strategic objectives, fostering scalable and sustainable outcomes.

(4) Lessons for future policymaking

The CICP's evolution underscores the importance of adaptive policymaking that integrates technological trends with institutional capacity. Future policies should prioritize three areas: strengthening market-driven mechanisms to complement government-led initiatives, enhancing policy operability through clearer guidelines and performance metrics, and fostering

**Table 13. Staticst of policy tool clauses in the two phases.**

| ID of Tools | | P1 | P2 | P3 | P4 | P5 | P6 | P7 | P8 | P9 | P10 | P11 | P12 | P13 | P14 | | | Number of Clauses |
|---|---|---|---|---|---|---|---|---|---|---|---|---|---|---|---|---|---|---|
| | | CEP (2010–2019) | | | | | | | | | | | | | | | | |
| SP Tools(T1) | T1-1 | 1 | | 1 | | 1 | 1 | | 1 | | | | | | | | | 5 |
| | T1-2 | | | 1 | 1 | 1 | | 1 | 1 | 1 | | 1 | 1 | | | | | 8 |
| | T1-3 | 1 | | 2 | | | 1 | 2 | 1 | 1 | | 1 | 1 | 1 | | | | 11 |
| | T1-4 | 2 | 1 | 1 | 1 | 4 | 1 | 1 | | | | 1 | | 1 | | | | 13 |
| | T1-5 | 1 | | 1 | | | 2 | 2 | 2 | 1 | | 1 | 3 | 1 | | | | 14 |
| DP Tool(T2) | T2-1 | 1 | | 1 | | | 1 | 1 | | 1 | | 1 | 1 | 1 | | | | 8 |
| | T2-2 | | | | | 1 | | | | | | | | | | | | 1 |
| | T2-3 | 2 | | 1 | | | | 1 | 1 | 1 | | 1 | 2 | 1 | | | | 10 |
| | T2-4 | 1 | | 1 | | | 1 | 1 | | | | | 2 | | | | | 6 |
| | T2-5 | | | | | | 1 | 1 | | | | 1 | | | | | | 3 |
| EP Tools(T3) | T3-1 | 1 | | | 1 | | 1 | 1 | 1 | 1 | | | 1 | 1 | | | | 8 |
| | T3-2 | 2 | 1 | | | 1 | 1 | 1 | 2 | 1 | | | | | | | | 9 |
| | T3-3 | 1 | | 3 | 1 | | 2 | 3 | 1 | 1 | 1 | 1 | 5 | | 1 | | | 20 |
| | T3-4 | 4 | | | | 1 | 1 | | | 1 | | | 1 | 1 | | | | 9 |
| | T3-5 | | | 1 | 1 | 1 | 1 | 1 | | 1 | | | | | | | | 6 |

| ID of Tools | | P15 | P16 | P17 | P18 | P19 | P20 | P21 | P22 | P23 | P24 | P25 | P26 | P27 | P28 | P29 | P30 | Number of Clauses |
|---|---|---|---|---|---|---|---|---|---|---|---|---|---|---|---|---|---|---|
| | | DPP (2020–2022) | | | | | | | | | | | | | | | | |
| SP Tools(T1) | T1-1 | | 1 | 1 | 1 | | | | 1 | | 1 | 1 | | | | | 2 | 8 |
| | T1-2 | 3 | 1 | 3 | 1 | | | 1 | 1 | 2 | 1 | 3 | 1 | 1 | | | 6 | 24 |
| | T1-3 | 1 | | 1 | | | | | 1 | 1 | 1 | 1 | | 1 | 1 | | 1 | 9 |
| | T1-4 | 1 | | 2 | 3 | | | | 2 | 2 | 3 | 1 | | 2 | | | 1 | 17 |
| | T1-5 | 1 | 1 | 2 | 2 | | | | 1 | 3 | 1 | | | 6 | 2 | | 2 | 21 |
| DP Tools(T2) | T2-1 | 3 | 1 | | 2 | | | | 2 | 1 | | 4 | | 3 | | 1 | 1 | 18 |
| | T2-2 | | | | | | | | | | | | | | | | | 0 |
| | T2-3 | 2 | 3 | 2 | 1 | 1 | | 1 | | | 1 | 2 | 1 | 2 | 1 | 1 | 4 | 22 |
| | T2-4 | 2 | 1 | 1 | 3 | | | | 2 | | 1 | 1 | | 1 | | | 1 | 13 |
| | T2-5 | | 1 | | | | | | | | | | | | 2 | | | 3 |
| EP Tools(T3) | T3-1 | 1 | 2 | 1 | | 1 | | | | | | 1 | | 3 | | | | 9 |
| | T3-2 | 1 | 2 | 1 | | 1 | | | 1 | 1 | 1 | 1 | | 1 | 2 | 1 | | 13 |
| | T3-3 | 1 | 3 | 4 | | | 5 | | 1 | 5 | 4 | 3 | | 4 | 1 | | 1 | 32 |
| | T3-4 | 1 | | 2 | 2 | | | | 1 | | | 1 | | | 1 | | | 8 |
| | T3-5 | 1 | 1 | 3 | | | 1 | | 1 | 2 | 1 | 1 | | 1 | | 1 | 1 | 14 |

innovation ecosystems that bridge academia, industry, and government. Additionally, maintaining flexibility to accommodate rapid technological changes—such as advancements in AI-driven construction or circular economy practices—will be critical.

However, the aforementioned research demonstrates that the formulation quality of CICP policies has not yet attained an optimal level, retaining measurable potential for enhancement. To further optimize CICP, it is essential to identify existing policy deficiencies through PMC surface analysis, particularly in dimensions corresponding to specific evaluation variables such as policy scope, effectiveness, and implementation mechanisms. By integrating these findings with an assessment of emerging trends in China's intelligent construction sector, policymakers can develop more targeted and effective strategies.

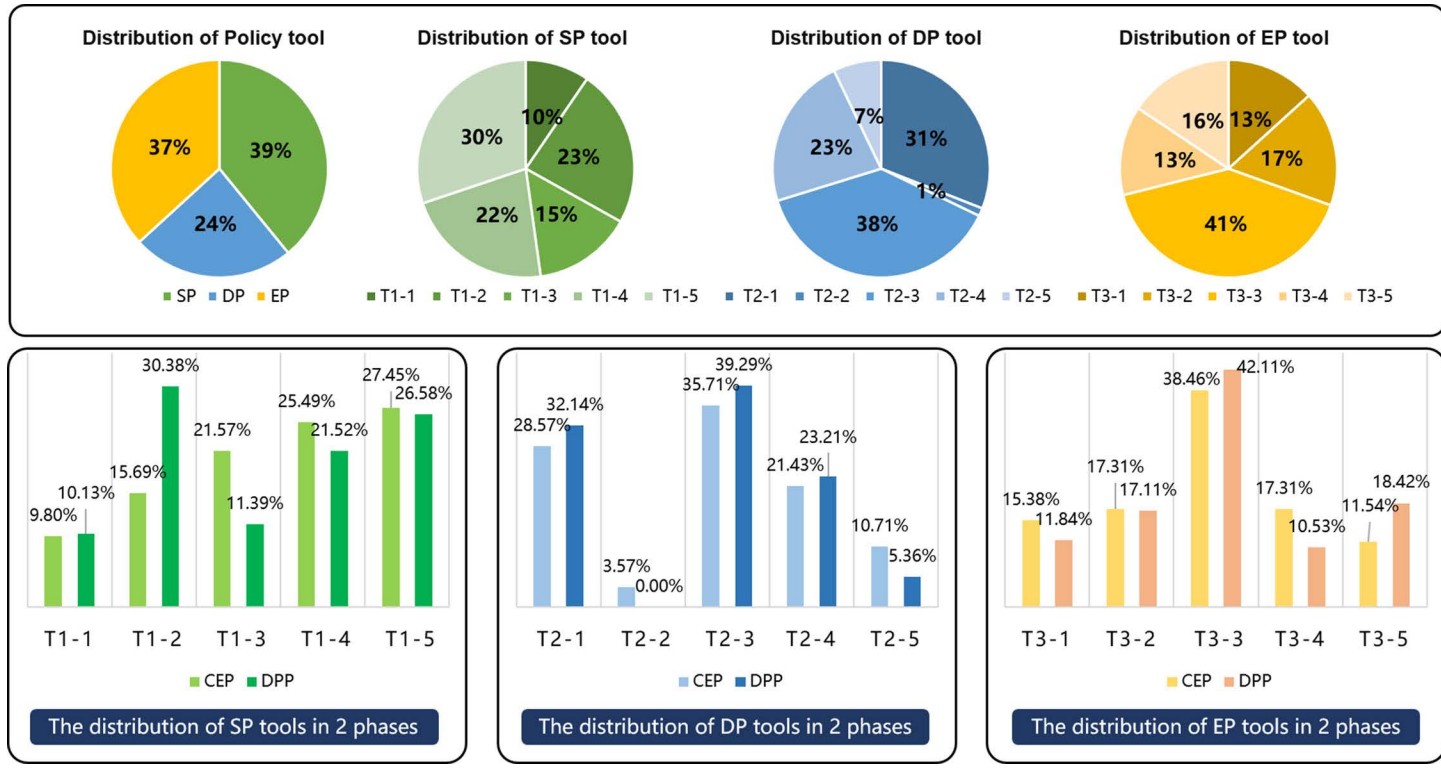

**Fig 13. Analysis of policy tool proportions in the two phases.**

## Discussions of policy gaps and development trends

**Policy gaps identification by constructing PMC surfaces.** Using PMC index values from Table 7, this research could create a PMC surface for each policy based on Formula (5), offering a more detailed visual analysis. Table 5 identifies four high-rated policies that do not reach the "Perfect" level. P15 and P25 are rated as "Excellent" with 8.36 and 8.27, respectively, while P27 and P17 achieve scores of 7.82 and 7.86.

PMC surfaces for these policies (Fig 15) highlight specific areas impacting their ratings, revealing gaps against the "Perfect" policy based on concavity index distribution. P15, guiding intelligent construction and construction industrialization, is comprehensive but needs refinement in "Policy Scope, " "Policy Perspective, " and "Issuing Institution. " P17, emphasizing high-quality growth in construction industrialization with intelligent construction as a driver, could improve in "Policy Scope, " "Policy Field, " "Policy Perspective, " and "Issuing Institution. " P25, focusing on carbon peak goals and reducing the construction industry's carbon footprint, suggests enhancements in "Policy Scope, " "Policy Perspective, " and "Issuing Institution. " P27, a strategic plan for the construction industry, highlights intelligent construction as a transformational focus but requires improvement in "Policy Scope, " "Policy Effectiveness, " "Incentive Constraints, " and "Issuing Institution. "

However, the analysis also indicates that the authority of the issuing institution(X8) impacts the ratings of the four policies. Given that policy formulation is led by SC and MHURD, with coordinated efforts across multiple ministries, there is already substantial authority within the single industry. Thus, this factor might not primarily influence the achievement of a perfect CICP formulation.

The findings above indicate that to achieve near-optimal policy formulation quality, CICP requires further refinements across four critical dimensions: expansion of policy scope, enhancement of policy effectiveness, broadening of regulatory perspectives, and diversification of incentive mechanisms. To establish actionable optimization guidelines and ensure

**CEP (2010–2019)**

| Group | SP Tools(T1): 33.33% (36 clauses) | | | | | DP Tools(T2): 20.37% (22 clauses) | | | | | EP Tools(T3): 46.30% (50 clauses) | | | | |
|---|---|---|---|---|---|---|---|---|---|---|---|---|---|---|---|
| | T1-1 | T1-2 | T1-3 | T1-4 | T1-5 | T2-1 | T2-2 | T2-3 | T2-4 | T2-5 | T3-1 | T3-2 | T3-3 | T3-4 | T3-5 |
| G1 |  |  |  | 1 |  |  | 1 |  |  |  | 1 | 3 |  |  |  |
| G2 | 1 | 1 |  | 1 |  |  | 1 |  |  | 1 | 1 | 2 | 2 | 1 |  |
| G3 | 1 | 2 |  | 2 |  |  | 1 | 2 |  |  | 3 | 4 | 8 |  | 1 |
| G4 |  | 1 |  |  | 1 |  |  | 1 |  | 1 | 1 | 3 | 1 |  |  |
| G5 |  | 1 | 1 | 1 | 1 | 1 | 1 | 1 |  |  |  | 2 | 1 |  |  |
| G6 |  | 1 |  | 1 | 1 |  | 1 | 1 |  |  |  | 3 | 1 |  |  |
| G7 | 3 | 2 | 2 | 5 | 6 | 2 |  | 2 | 4 | 1 |  | 1 | 6 | 2 | 3 |
| % | 13.89% | 22.22% | 8.33% | 30.56% | 25.00% | 13.64% | 22.73% | 31.82% | 18.18% | 13.63% | 12.00% | 36.00% | 38.00% | 6.00% | 8.00% |

**DPP (2020–2022)**

| Group | SP Tools(T1): 43.48% (110 clauses) | | | | | DP Tools(T2): 31.62% (80 clauses) | | | | | EP Tools(T3): 24.90% (63 clauses) | | | | |
|---|---|---|---|---|---|---|---|---|---|---|---|---|---|---|---|
| | T1-1 | T1-2 | T1-3 | T1-4 | T1-5 | T2-1 | T2-2 | T2-3 | T2-4 | T2-5 | T3-1 | T3-2 | T3-3 | T3-4 | T3-5 |
| G1 | 1 | 9 | 1 | 4 | 3 | 6 |  | 7 | 7 |  | 4 | 3 | 6 | 3 | 4 |
| G2 |  | 15 |  | 5 | 1 | 2 |  | 6 | 3 |  |  | 1 | 2 |  | 1 |
| G3 |  | 10 |  | 3 | 1 | 2 |  | 7 | 3 |  |  |  | 9 |  | 1 |
| G4 |  | 5 |  | 2 | 3 | 2 |  | 5 | 2 | 1 | 2 |  | 11 |  |  |
| G5 | 1 | 8 |  | 5 | 2 | 2 |  | 3 | 6 | 1 |  |  | 3 |  |  |
| G6 |  | 9 |  | 5 | 1 | 2 |  | 6 | 5 |  |  |  | 2 |  |  |
| G7 | 3 | 1 |  | 2 | 10 | 1 |  | 1 |  |  |  | 2 | 5 | 2 | 2 |
| % | 4.55% | 51.82% | 0.09% | 23.64% | 19.90% | 21.25% | 0.00% | 43.75% | 32.50% | 2.50% | 9.52% | 9.52% | 60.32% | 7.94% | 12.70% |

**Fig 14. Coupling analysis of policy goals and tools based on the statics of clauses.**

the temporal relevance and sustainability of future policies, it is imperative to align CICP's developmental trajectory with cutting-edge research frontiers in China's intelligent construction field.

**Development trends discussion with the research frontier.** This research offers integrating research frontier insights for a deeper analysis. A thorough search for "intelligent construction" was conducted in the CNKI, China's preeminent academic literature database. After data cleaning and selection, the research analyzed seven hundred ninety-two literature samples (2021–2023). The keyword co-occurrence network was created using Vosviewer, a bibliometric tool, revealing research frontiers and hotspots in China's intelligent construction field (see Table 14). The results show that China's intelligent construction sector is undergoing a transformative phase driven by technological convergence and policy innovation. At its core, the integration of BIM (Building Information Modeling, FQ = 165) with artificial intelligence (FQ = 32) and digital twins (FQ = 29) has established a technological triad that reshapes traditional construction paradigms. This synergy fuels advancements in smart construction sites (FQ = 41) and prefabricated systems (FQ = 66), where robotic applications (FQ = 26) and 3D printing (FQ = 8) are increasingly bridging the gap between digital design and physical execution. Notably, the rising emphasis on "digital transformation" (FQ = 16) reflects China's strategic alignment with global Industry 4.0 frameworks while adapting them to its unique urbanization challenges.

Parallel to technological breakthroughs, the research frontier highlights systemic industrial upgrading through "construction industrialization" (FQ = 88) and "high-quality development" (FQ = 31). Emerging priorities such as full

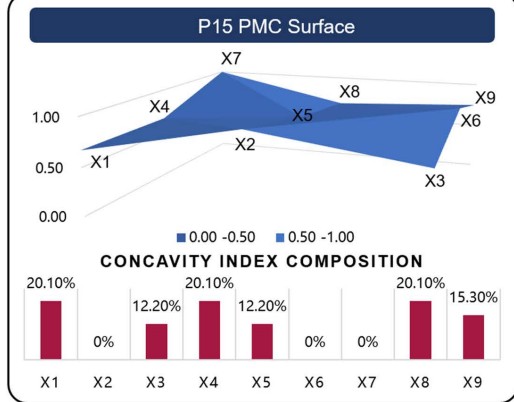
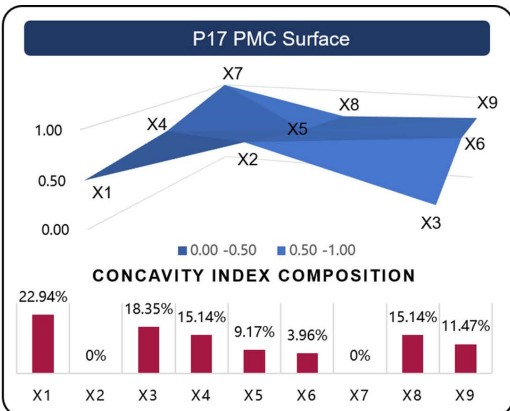
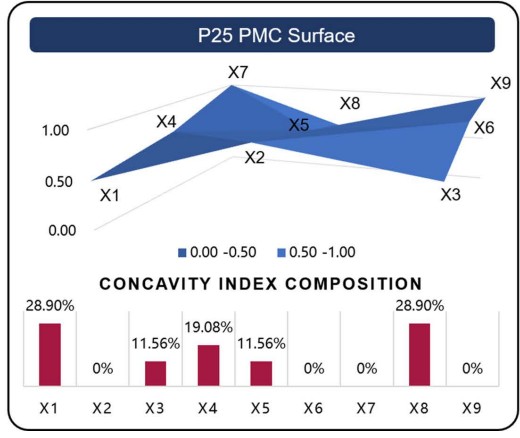
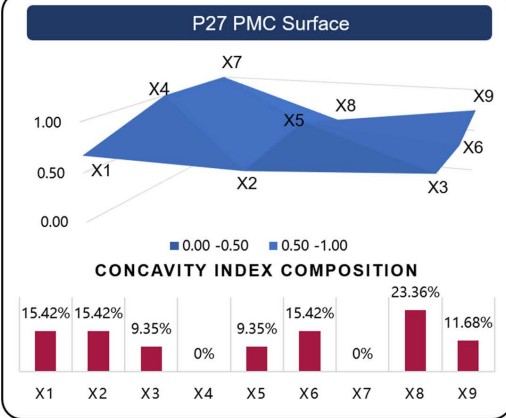

**Fig 15. PMC surfaces and concavity index distribution for four policies.**

lifecycle management (FQ = 15) and green construction (FQ = 22) underscore a dual focus on process optimization and sustainability. However, the fragmented adoption of collaborative development models (FQ = 39) and persistent gaps in cross-industry data integration (implied by low "standardization" FQ = 6) signal unresolved challenges in achieving true industry-wide digital cohesion.

Policy frameworks play a pivotal yet evolving role. While initiatives like "new construction industrialization" (FQ = 22) demonstrate effective alignment with technological goals, the underdeveloped financial incentives for SMEs (implied by FQ = 5) and mismatches in talent cultivation (FQ = 30) reveal critical gaps in policy instrument design. This disconnect is further evidenced by the PMC index analysis, which identifies inconsistencies in policy coherence despite post-2020 improvements.

Looking ahead, three interconnected trajectories will define China's intelligent construction evolution:

Firstly, Deepening human-technology symbiosis through AI-augmented workforce training and human-robot collaboration systems, addressing the "talent training model" (FQ = 30) challenges while scaling up computer vision (FQ = 4) and IoT-enabled safety solutions (FQ = 23).

Secondly, Policy-innovation feedback loops translate research insights into adaptive governance, particularly in standardizing BIM-Digital Twin integration and creating dynamic incentive mechanisms for SME participation.

Finally, Ecosystemic sustainability via closed-loop material flows powered by robotic prefabrication and blockchain-tracked supply chains, operationalizing "green construction" (FQ = 22) concepts at scale.

**Table 14. Cluster statistics of high-frequency keywords.**

| Cluster1 | FQ | Cluster2 | FQ | Cluster3 | FQ | Cluster4 | FQ | Cluster 5 | FQ |
|---|---|---|---|---|---|---|---|---|---|
| Construction Industry | 88 | Intelligent Construction | 359 | Construction Industrialization | 88 | Smart Construction Site | 41 | Prefabricated Construction | 66 |
| High-Quality Development | 31 | Talent Training Model | 30 | Collaborative Development | 39 | Internet of Things | 23 | Smart Construction | 26 |
| New Construction Industrialization | 22 | Civil Engineering Construction | 19 | Digital Twin | 29 | Full Lifecycle | 15 | Steel Structure Construction | 10 |
| Green Construction | 22 | Curriculum System | 18 | Lean Construction | 11 | Informatization | 10 | BIM Technology Application | 8 |
| MHURD | 15 | New Engineering Disciplines | 15 | Industrialization | 9 | General Contracting of Engineering | 10 | Large Steel Structures | 8 |
| Construction Technology | 12 | Teaching Reform | 9 | Intelligent Design | 6 | Full Industry Chain | 9 | Green Buildings | 6 |
| Construction Method | 9 | Practical Teaching | 7 | Industry 4.0 | 5 | Cloud Computing | 8 | Smart Management Platform | 5 |
| Full Process Management | 7 | Professional Construction | 6 | "Opinion" | 4 | CIM | 6 | Information Technology | 4 |
| Construction Industry Association | 6 | Application | 6 | Construction Industry Informatization | 4 | Big Data | 5 | Collaborative Management | 4 |
| Survey and Design Industry | 5 | Industry-Education Integration | 5 | New Momentum | 4 | Current Development | 4 | Construction Technology | 4 |
| Construction Enterprises | 5 | Engineering Cost | 5 | Intelligent Management | 4 | Basic Platform | 4 | Construct Progress | 4 |
| Urban Renewal | 4 | Construction Engineering | 4 | Mechanical Equipment | 4 | Real Name System | 4 | Evaluation System | 4 |
| Construction Industry Development Planning | 4 | Innovation and Entrepreneurship | 4 | | | | | | |
| Intelligent Technology | 4 | | | | | | | | |
| Science and Technology Innovation | 4 | | | | | | | | |

| Cluster 6 | FQ | Cluster7 | FQ | Cluster8 | FQ | Cluster 9 | FQ |
|---|---|---|---|---|---|---|---|
| Artificial Intelligence | 32 | BIM | 165 | Construction Robots | 26 | Intelligent Equipment | 16 |
| Digital Construction | 25 | Digitalization | 17 | Digital Transformation | 16 | Intelligent Manufacturing | 14 |
| 3D Printing | 8 | Engineering Management | 16 | Construction Industry Enterprises | 9 | PC Components | 8 |
| Intelligent Robots | 8 | Intelligentization | 7 | Influencing Factors | 7 | Building Intelligence | 7 |
| Parametric Design | 7 | Construction Management | 6 | Construction Industry Internet | 6 | Construction Site | 4 |
| Architectural Design | 6 | Green Construction | 6 | Smart Operation and Maintenance | 6 | | |
| Digital Design | 6 | Automation | 5 | | | | |
| Intelligent Detection | 4 | Visualization | 4 | | | | |
| Robotic Arm | 4 | | | | | | |
| Deep Leaning | 4 | | | | | | |
| Computer Vision | 4 | | | | | | |

The path forward demands tighter coupling between academic research, industry implementation, and policy experimentation. As China advances toward its 14th Five-Year Plan targets, success will hinge on transforming current technological leadership in areas like parametric design (FQ = 7) and smart management platforms (FQ = 5) into comprehensive, policy-supported ecosystems that redefine global construction standards.

## Discussions and recommendations on the optimization of CICP

The evolution of CICP reveals a dynamic interplay between technological innovation and institutional adaptation, yet persistent gaps in policy design and implementation necessitate targeted optimization. Building on the identified research frontiers and PMC surface analysis, four strategic dimensions emerge as critical for enhancing policy coherence and operational impact.

Expanding the nature of policy requires transcending the current government-centric paradigm to embrace hybrid governance models that integrate market-driven mechanisms. While existing policies emphasize top-down directives—such as BIM adoption targets and industrialization roadmaps—the underdeveloped financial incentives for SMEs (implied by low FQ = 5 in "standardization") highlight a systemic imbalance. To address this, policies should cultivate public-private innovation ecosystems, leveraging blockchain-enabled procurement platforms and outcome-based contracting frameworks. For instance, embedding performance-linked tax rebates for enterprises adopting closed-loop material flows or AI-driven carbon accounting systems could align fiscal tools with sustainability goals, as advocated in the "green construction" (FQ = 22) research frontier.

Strengthening policy effectiveness demands a shift from broad strategic statements to granular, metrics-driven implementation. The PMC analysis underscores deficiencies in "policy effectiveness" (notably in P27) and "incentive constraints," reflecting a disconnect between high-level objectives like "high-quality development" (FQ = 31) and on-ground execution. An actionable pathway involves establishing a dynamic monitoring framework that quantifies progress in key areas such as BIM-Digital Twin integration (FQ = 165 and 29, respectively) and robotic prefabrication (FQ = 66). This could include mandatory lifecycle carbon audits for state-funded projects or real-time data-sharing mandates across the "full industry chain" (FQ = 9), enforced through interoperable digital platforms.

Broadening policy perspectives entails dismantling silos between technological advancement and socio-industrial transformation. Current policies, while progressive in areas like "smart construction sites" (FQ = 41), inadequately address cross-sectoral challenges such as talent mismatches ("talent training model" FQ = 30) and fragmented data governance. A holistic approach would involve co-designing policies with academia and industry to align "new engineering disciplines" (FQ = 15) with emerging skill demands—for example, integrating AI-augmented design training modules into vocational curricula and creating cross-ministerial task forces to standardize IoT protocols (FQ = 23) for construction robotics.

Diversifying incentives must move beyond traditional subsidies toward multidimensional motivators that catalyze systemic change. The PMC surface analysis of P15 and P25 reveals underutilized potential in "policy scope" and "incentive mechanisms," correlating with the research frontier's emphasis on "collaborative development" (FQ = 39). Innovative solutions could include tiered R&D tax credits for enterprises adopting "Industry 4.0" (FQ = 5) technologies, coupled with risk-sharing funds for piloting AI-driven project management systems. Simultaneously, introducing "green bond" mechanisms specifically for intelligent construction projects could channel private capital into sustainable infrastructure, operationalizing the "dual carbon" goals embedded in policies like P25.

Ultimately, optimizing CICP demands a synergistic recalibration of these dimensions, anchored in continuous feedback loops between policy experimentation and research insights. By embedding adaptive governance structures—such as AI-powered policy simulators to forecast BIM adoption barriers or decentralized autonomous organizations (DAOs) for stakeholder collaboration—the CICP framework can evolve from incremental improvements to transformative, future-proof systems. This alignment with both technological frontiers and institutional realities will solidify China's leadership in redefining global construction paradigms.

## Conclusions

### Summary and the findings of the research

This study provides a systematic evaluation of CICP from 2010 to 2022, utilizing the PMC index model to assess the formulation quality, evolution, and effectiveness of 30 national policies. The findings reveal that:

(1) CICP demonstrates satisfactory overall quality, with an average PMC score of 6.93 ("Acceptable"). The policies exhibit distinct phased development characteristics, divided into the Cultivation and Exploration Phase (CEP: 2010–2019) and the Development and Promotion Phase (DPP: 2020–2022). The CEP focused on foundational digital infrastructure and technological innovation, such as BIM and IoT, but faced challenges in practical application and policy continuity. In contrast, the DPP achieved higher policy quality, with an average PMC score of 7.03 ("Positive"), and saw the emergence of "Excellent" policies like P15 and P25. These policies emphasized industrialization, digitalization, and sustainability, driving rapid advancements in intelligent construction.

(2) The evolution of CICP marks a transition from fragmented, exploratory strategies to integrated, goal-oriented frameworks. Key factors driving this shift include technological advancements, accumulated implementation experience, and heightened global competition. The DPP's effectiveness stems from its ability to align policy tools with strategic goals, fostering scalable and sustainable outcomes. Lessons for future policymaking highlight the importance of adaptive governance, integrating technological trends with institutional capacity, and fostering innovation ecosystems that bridge academia, industry, and government.

(3) The study identifies persistent gaps in policy scope, effectiveness, and incentive mechanisms. The PMC surface analysis of high-rated policies (P15, P17, P25, and P27) reveals deficiencies in areas such as policy scope, regulatory perspectives, and implementation frameworks. These gaps underscore the need for further refinement to achieve near-optimal policy formulation quality. Additionally, the research frontier analysis indicates that China's intelligent construction sector is undergoing a transformative phase driven by technological convergence and policy innovation, with emerging priorities such as full lifecycle management and green construction. Future trends suggest a deepening of human-technology symbiosis, policy-innovation feedback loops, and ecosystemic sustainability.

### Recommendations about CICP

(1) Government policy optimization framework

The government should adopt a holistic approach to policy formulation, integrating top-down strategic planning with bottom-up market incentives. This includes fostering public-private partnerships (PPPs) and leveraging blockchain-enabled transparency in procurement. Adaptive governance structures, such as AI-powered policy simulators, should be established to forecast implementation challenges and optimize resource allocation.

(2) Addressing policy gaps

Expanding Policy Scope. Policies should transcend the current government-centric paradigm by integrating market-driven mechanisms. For instance, performance-linked tax rebates could be introduced for enterprises adopting sustainable practices, such as closed-loop material flows or AI-driven carbon accounting systems. Hybrid governance models, including outcome-based contracting, should be developed to incentivize private sector participation in intelligent construction projects.

Strengthening Policy Effectiveness. To bridge the gap between high-level objectives and on-ground execution, policies should adopt granular, metrics-driven implementation frameworks. A dynamic monitoring system should be established to quantify progress in key areas, such as BIM-Digital Twin integration and robotic prefabrication. Mandatory lifecycle carbon

audits for state-funded projects and real-time data-sharing mandates across stakeholders via interoperable platforms could enhance accountability and transparency.

Broadening Policy Perspectives. Policies must address cross-sectoral challenges by dismantling silos between technological advancement and socio-industrial transformation. Cross-ministerial task forces should be formed to standardize IoT protocols for construction robotics and align vocational training with emerging skill demands, such as AI-augmented design. Additionally, global alignment with ISO standards for intelligent construction should be pursued while tailoring frameworks to China's unique urbanization challenges.

Diversifying Incentives. Moving beyond traditional subsidies, policies should introduce multidimensional motivators that catalyze systemic change. Tiered R&D tax credits could be offered to enterprises adopting Industry 4.0 technologies, such as 3D printing and robotic automation. Green financing mechanisms, such as "intelligent construction bonds," should be launched to attract private capital into sustainable infrastructure projects.

(3) Future optimization directions

Future policies should focus on deepening human-technology symbiosis through AI-augmented workforce training programs and human-robot collaboration systems. Ecosystemic sustainability should be promoted by operationalizing circular economy principles, such as blockchain-tracked supply chains and robotic prefabrication. Strengthening ties between academia and policymakers will be critical to translating frontier research (e.g., digital twins, smart materials) into actionable policies.

Implementing these recommendations would enable to address current policy gaps, align CICP with global technological frontiers, and solidify its leadership in intelligent construction, ensuring sustainable and high-quality development.

## Strengths and limitations

This paper makes several unique contributions to the field of CICP research. First, it provides scientific insights into the evolution, structural characteristics, and formulation quality of CICP, offering data-based suggestions for policy optimization. Second, the research introduces an innovative dynamic and comparative approach that incorporates a temporal dimension—a key element often overlooked in traditional policy evaluations. By considering the time-sensitive nature of policy formulation and implementation, this study provides a more relevant and nuanced framework for assessing and improving policies over time. The PMC index model used in this study offers a fresh perspective on policy evaluation, enabling policymakers to track the progression and effectiveness of intelligent construction policies quantitatively. This methodological approach not only fills a gap in existing literature but also enhances the reliability and objectivity of policy assessments, setting a new standard for future research.

However, methodological diversity characterizes policy evaluation research. There exists no universally applicable criterion for selecting policy evaluation variables across different studies, as variable determination fundamentally depends on specific research contexts, theoretical frameworks, and policy implementation environments. Critically, this research focuses on national policies in China, emphasizing macro-level features with detailed exploration. While this approach offers valuable insights into the broader policy landscape, it also comes with some limitations. The central government's strong directives have influenced regional authorities to introduce a range of policies aimed at stimulating intelligent construction initiatives in recent years. However, the analysis of national policies, though comprehensive, may overlook the practical implementation challenges at the local level. For example, regional variations in economic conditions, institutional capacities, and enforcement mechanisms can result in differing outcomes despite alignment with national directives.

Moreover, the reliance on textual data to assess policy content may fail to capture the nuanced, qualitative aspects of policy execution, such as local adaptation processes and the real-world challenges faced by stakeholders. These limitations should be considered when interpreting the results of this study. Recognizing the creation, implementation, and

performance of these policies at both the national and regional levels is crucial, as local policies aligned with national directives can stimulate diverse and new research inquiries within the CICP domain.

Future research could address these gaps by incorporating qualitative methods or case studies, including:

Longitudinal Studies. Future studies could incorporate longitudinal research to explore the long-term impacts of intelligent construction policies, assessing how policies evolve and their sustained effects over time.

Qualitative Research. Further research could involve qualitative studies that collect feedback from stakeholders in the construction industry. This would provide deeper insights into the real-world challenges of policy implementation and complement the quantitative analysis presented in this paper.

Cross-Regional Comparisons. Expanding research to compare policies across different regions could reveal how local contexts influence policy outcomes, offering valuable lessons for scaling up intelligent construction initiatives nationwide.

These future research directions will help build upon the findings of this paper, providing a more comprehensive understanding of how to optimize intelligent construction policies in a rapidly evolving field.

## Supporting information

**S1 File.** S1 Appendix. the thirty CICP texts. S2 Appendix. datasets for tables 1–14. S3 Appendix. datasets for figures 3–15. S4 Appendix. academic literature data of intelligent construction (CNKI). S5 Appendix. High-Frequency Keywords of intelligent construction.
(ZIP)

## Acknowledgments

The authors thank all colleagues who participated in the paper's research and writing/reviewing.

## Author contributions

**Conceptualization:** Xiongquan Ou.

**Data curation:** Xiongquan Ou.

**Formal analysis:** Xiongquan Ou, Ming Ma.

**Investigation:** Xiongquan Ou, Wei Wang.

**Methodology:** Xiongquan Ou.

**Project administration:** Ming Ma.

**Resources:** Xiongquan Ou.

**Software:** Wei Wang.

**Supervision:** Ming Ma.

**Validation:** Xiongquan Ou, Ming Ma.

**Visualization:** Wei Wang.

**Writing – original draft:** Xiongquan Ou.

**Writing – review & editing:** Xiongquan Ou, Ming Ma, Wei Wang.

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
