## [Editor Report · Decision Letter 0]

Dear Dr. Ma,

Thank you for submitting your manuscript to PLOS ONE. After careful consideration, we feel that it has merit but does not fully meet PLOS ONE’s publication criteria as it currently stands. Therefore, we invite you to submit a revised version of the manuscript that addresses the points raised during the review process.

Revise and Resubmit.

This paper provided a scientific analysis and evaluation of China's Intelligent Construction Policies (CICP) using text mining, bibliometric analysis, and the Policy Modeling Consistency (PMC). The authors identified several key challenges in the in-depth promotion of CICP, including imperfections in policy guarantees and a lack of comprehensive content in the policies using the PMC index model to quantitatively analyze and evaluate the formulation quality of CICP, as well as review and discuss the primary features of CICP from a dynamic analysis framework. This paper highlighted the importance of developing a multi-dimensional analytical framework for quantitatively assessing CICP to identify the policies' features and understand the outcomes in policy formulation given most if the existing research on CICP is predominantly based on empirical feedback or qualitative analysis, with a lack of quantitative evaluation of policy texts.

Here are some my key comments and suggestions:

1.     Lines 42-45, authors used a few examples to introduce the technologies. However, not all of them are trivial or clear to the readers why they are considered "core technologies" and What role they play in construction.

2.     Another key contribution of the paper that the authors claimed is to fill the gap between existing literatures. Lines 62-64, authors mentioned a lack of quantitative analysis in policy evaluation but doesn’t explain the significance of filling this gap. Please Expand on why this gap matters. For example, explain how a quantitative approach would improve the accuracy of policy assessments or how it could impact future policy decisions.

3.     The literature review session, lines 98-124, can be improved. The literature review feels like a collection of studies without a clear thematic grouping. I would suggest that group the studies into clear thematic sub-sections (e.g., technological innovation in construction, policy evolution in construction). This will help readers follow the progression of ideas.

4.     The authors employed PMC Model in the study and assumed readers are familiar with the PMC index model, making it hard for a general reader to follow. I would suggest the authors to offer a simplified explanation or a step-by-step example of how the PMC index model is applied, somewhere in between lines 204-244, possibly with a small sample policy to illustrate the process. Figures 1 and 2 could be accompanied by a brief walkthrough.

5.     The selection of the variables is the key of this study. The paper listed the variables but doesn’t explain why they are relevant to evaluating intelligent construction policies. Would the author be able to provide the model specification? At least, provide a rationale for selecting specific variables. For instance, explain why certain policy aspects (e.g., effectiveness, macro or micro perspective) are crucial in evaluating intelligent construction policies.

6.     Around line 217, the authors mentioned briefly about the limitations, but did not discuss in depth. Please add a more detailed limitations section. For instance, mention how the quantitative nature of the PMC model might miss qualitative nuances, or how relying on textual data might overlook the practical implementation of policies in different regions.

7.     Lines 333-348, authors summarized the results. The discussion scale was limited, not in depth. Please provide a more deep insight into the results. For example, explain why the shift between phases is significant. For instance, how did policy changes impact construction practices? Provide examples of how policies in Phase 2 (Development and Promotion) improved upon issues from Phase 1 (Cultivation and Exploration). Please also provide more details about the specific factors that led to the policy shift and how these changes addressed earlier issues. Discuss why Phase 2 policies are more effective and what lessons can be learned for future policymaking.

8.     The most exciting section of the paper should be the recommendation section, around line 533. The recommendations given by the authors were somewhat broad and could benefit from more actionable suggestions. Instead of general statements, provide concrete actions that policymakers can take. For example, suggest specific areas where policy improvements can focus, like integrating AI more effectively into construction planning or increasing financial incentives for small and medium enterprises in construction. I would also suggest the authors, after presenting each recommendation, please link it back to a specific finding from the study. For example, if a specific policy scored lower on consistency, suggest changes to improve this aspect in future policies.

9.     I would suggest the authors to highlight the unique contributions of the paper. For example, emphasize how the PMC index provides a new perspective on policy evaluation or how the findings reveal important gaps in China’s current intelligent construction policies. And then point out specific suggestions for future research. For instance, suggest conducting longitudinal studies to track the long-term impact of policies or qualitative studies that involve feedback from stakeholders in the construction industry.

We look forward to receiving your revised manuscript.

Kind regards,

Ai Ren

Guest Editor

PLOS ONE

Journal Requirements:

This research was funded by China Postdoctoral Science Foundation (grant number 2022M722401). 

3. We note that your Data Availability Statement is currently as follows: [All relevant data are within the manuscript and its Supporting Information files.
---

## [Author Response · Author response to Decision Letter 1]

22 Mar 2025

Response to Reviewer Comments

Manuscript ID: PONE-D-24-31927

Title: Research on quantitative evaluation of China's Intelligent Construction Policy (CICP) based on the Integration of PMC index model and multi-dimensional analytical framework

Dear Reviewer,

We sincerely appreciate your rigorous evaluation and insightful feedback on our manuscript. Your expertise has been invaluable in refining this work, and we are deeply grateful for the time and intellectual rigor you devoted to the review process. In response to your critiques, we formed a dedicated revision team to systematically address each point raised:

Comment 1: Lines 42-45, authors used a few examples to introduce the technologies. However, not all of them are trivial or clear to the readers why they are considered "core technologies" and What role they play in construction.

Response: We appreciate your valuable feedback. To address this concern, we have revised the introduction of the core technologies in the manuscript to provide clearer explanations regarding their significance and roles in the construction industry. Specifically, we have elaborated on the relevance of each technology in the context of modern construction practices, highlighting how each one contributes to improving efficiency, sustainability, or safety. Additionally, we have clarified why these particular technologies are considered "core" in the field. These clarifications can be found on lines 42-51 of the revised manuscript.

Comment 2: Another key contribution of the paper that the authors claimed is to fill the gap between existing literatures. Lines 62-64, authors mentioned a lack of quantitative analysis in policy evaluation but doesn’t explain the significance of filling this gap. Please Expand on why this gap matters. For example, explain how a quantitative approach would improve the accuracy of policy assessments or how it could impact future policy decisions.

Response: Thank your for pointing out the need for further elaboration on the significance of addressing this gap in the literature. We agree that a more detailed explanation would strengthen the argument for the importance of this gap. In response, we have expanded the discussion in the revised manuscript to emphasize how the lack of quantitative analysis in policy evaluation limits the ability to assess the true impact of policies. Specifically, we have explained that a quantitative approach can provide more robust and objective insights into policy effectiveness, helping to identify patterns, assess outcomes with greater precision, and inform better decision-making. Furthermore, we have discussed how such an data-driven approach can enhance the credibility of policy recommendations by offering empirical evidence that supports or challenges policy assumptions. We believe this is particularly important in the development process of CICP, where accurate assessments are crucial for long-term success. These clarifications can be found on lines 66-75 of the revised manuscript.

Comment 3: The literature review session, lines 98-124, can be improved. The literature review feels like a collection of studies without a clear thematic grouping. I would suggest that group the studies into clear thematic sub-sections (e.g., technological innovation in construction, policy evolution in construction). This will help readers follow the progression of ideas.

Response: We sincerely appreciate the reviewer’s suggestion to improve the structure of the literature review. We agree that organizing the studies thematically will help to enhance the clarity and flow of ideas. In response to this comment, we have reorganized the literature review into thematic sub-sections as recommended. The new structure includes the following sub-sections:

Themes of the existing studies, including thematic areas such as Technology Application and Infrastructure construction, Environment Improvement and Platform Support, Standard-setting and Management Innovation, Talent Development and System Optimization. This classification also points out the main focus on CICP evaluation and optimization in current.

Gaps in Current Research, noting that most of the policy recommendations made by existing studies were based on expert experience and lacked empirical evidence, and puts forward the necessity and importance of introducing quantitative analysis method to evaluate CICP scientifically.

Advances in Quantitative Policy Analysis, presenting a series of research cases, and to demonstrate the feasibility of scientific and systematic evaluation and optimization of CICP based on multi-dimensional research framework and quantitative analysis methods.

This grouping provides a clearer progression of ideas, making it easier for readers to follow the discussion and better understanding of the progress of existing CICP research and the reasons for quantitative analysis and evaluation of CICP. We believe these changes significantly improve the overall readability of the literature review. These clarifications can be found on lines 105-142 of the revised manuscript.

Comment 4: The authors employed PMC Model in the study and assumed readers are familiar with the PMC index model, making it hard for a general reader to follow. I would suggest the authors to offer a simplified explanation or a step-by-step example of how the PMC index model is applied, somewhere in between lines 204-244, possibly with a small sample policy to illustrate the process. Figures 1 and 2 could be accompanied by a brief walkthrough.

Response: We appreciate the reviewer’s helpful suggestion to make the application of the PMC Model more accessible to a broader audience. We agree that a more detailed explanation will enhance the clarity of the manuscript. In addition, we are aware that the presentation of the research methodology in the original text is complicated, which is not conducive to the reader's understanding. In response, we have reorganized the presentation of methodology and divided it into three parts:

Methodological Framework: This section explains that the analytical framework constructed in this paper relies on text mining and data visualization techniques, and covers the objective evaluation, comprehensive discussion, and optimization suggestions of CICP. In our analytical framework, the research work will be carried out according to five parts of advanced work, including the policy evaluation based on PMC Index Model and the dynamic analysis of policy evolution, status, and trend, offering data-driven recommendations for the optimization of CICP. This explanation, which is now included between lines 220-225, clearly outlines the process. We also revised Figures 1 to provide a clearer explanation, helping readers understand the key concepts being visualized.

Research Methods and Working Procedures: This section describes how we are excuting the five parts of advanced work in the analytical framework described above,and the methods and tools uesd in the process. At the same time, we also point out the final goals and conclutions pursued by these parts of the research work. This explanation, which is now included between lines 230-236, clearly outlines the methods and procedures. We also revised Figures 2 to provide a clearer explanation. Especially, we introduce the application method of the PMC index model in detail. We have added a simplified, step-by-step walkthrough of the PMC index model’s application, including an illustrative example using a small sample policy. This explanation, which is now included between lines 239-365, clearly outlines the process, showing how the model is applied and how the results are interpreted. We believe these additions will make the methodology more comprehensible to readers unfamiliar with the PMC Model.

Data sources and sample selection: This section explains where the data was obtained and processed, and details the final sample of the study. We have made no changes to this part.

We believe that the revised formulation will allow readers to understand the research methods used in this article more clearly and directly. All these modifications are included in the revised manuscript on pages 6-11.

Comment 5: The selection of the variables is the key of this study. The paper listed the variables but doesn’t explain why they are relevant to evaluating intelligent construction policies. Would the author be able to provide the model specification? At least, provide a rationale for selecting specific variables. For instance, explain why certain policy aspects (e.g., effectiveness, macro or micro perspective) are crucial in evaluating intelligent construction policies.

Response: We thank the reviewer for pointing out the need to clarify the rationale behind the selection of variables. We fully agree that explaining the relevance of these variables is essential for understanding their role in evaluating intelligent construction policies. In response, we have added a detailed explanation in the revised manuscript regarding the selection of each variable, focusing on their relevance to the evaluation of CICP. These clarifications can be found on lines 387-433 of the revised manuscript.

In addition, We further add an explanation of how primary and secondary variables are selected in the methodology section. In general practice, the primary variables are employed to assess the consistency of policies. These variables exhibit minimal variation in their configuration across different studies, allowing researchers to reference classical settings from existing applications of PMC model. Secondary variables, operating within the scope of primary variables, are particularly instrumental in identifying deficiencies and shortcomings in policy formulation. The selection of corresponding variables typically derives from a systematic synthesis of research findings within the subject's domain, complemented by empirical feedback from practical implementation experiences. (see lines254-266 of the revised manuscript)

Comment 6: Around line 217, the authors mentioned briefly about the limitations, but did not discuss in depth. Please add a more detailed limitations section. For instance, mention how the quantitative nature of the PMC model might miss qualitative nuances, or how relying on textual data might overlook the practical implementation of policies in different regions.

Response: We appreciate the reviewer’s suggestion to provide a more detailed discussion of the limitations in our study. We agree that a thorough consideration of the limitations will offer readers a more balanced view of the methodology and findings. In response, we have expanded the limitations section of the manuscript to address the points raised by the reviewer, as well as additional considerations. Specifically, we now discuss the following limitations in more detail:

Quantitative Nature of the PMC Model: While the PMC index model provides a robust quantitative framework for policy evaluation, its reliance on numerical data can overlook certain qualitative nuances, such as the context-specific factors or the subjective experiences of stakeholders. However, methodological diversity characterizes policy evaluation research.There exists no universally applicable criterion for selecting policy evaluation variables across different studies, as variable determination fundamentally depends on specific research contexts, theoretical frameworks, and policy implementation environments. We also acknowledge that the model may not capture the full complexity of policy impacts, particularly those that are difficult to quantify, such as cultural or societal effects.

Relying on Textual Data: The study's reliance on textual data to assess policies may overlook the practical, real-world implementation challenges faced by policymakers in different regions. The textual data often reflects the intended goals and objectives of policies, but may not fully account for variations in local conditions, such as regional economic contexts, regulatory environments, or capacity for policy enforcement.

We also address potential limitations related to sample size, data availability, and generalizability of results, and provide suggestions for how future research could mitigate these challenges.

These points have been incorporated into the revised manuscript in lines 956-986, and we believe that this expanded discussion offers a more comprehensive understanding of the study's limitations.

Comment 7: Lines 333-348, authors summarized the results. The discussion scale was limited, not in depth. Please provide a more deep insight into the results. For example, explain why the shift between phases is significant. For instance, how did policy changes impact construction practices? Provide examples of how policies in Phase 2 (Development and Promotion) improved upon issues from Phase 1 (Cultivation and Exploration). Please also provide more details about the specific factors that led to the policy shift and how these changes addressed earlier issues. Discuss why Phase 2 policies are more effective and what lessons can be learned for future policymaking.

Response: We thank the reviewer for the insightful feedback and for encouraging us to delve deeper into the discussion of the results. In response, we have significantly expanded the discussion to provide more in-depth analysis and insight into the shift between the phases and the specific impacts of policy changes. Below are the key additions:

Significance of the Shift Between Phases and Impact of Policy Changes on Construction Practices

Our comprehensive analysis of the transition from the CEP to the DPP yields several critical insights:

Firstly, the evaluation of 30 CICP texts using the PMC model, combined with comparative data analysis, reveals that the introduction of “P15” around 2020 represents a pivotal milestone in CICP's evolution. Statistical evidence demonstrates a marked improvement in policy formulation quality post-2020, indicating the commencement of a new developmental phase in CICP framework.( see lines 471-491 of the revised manuscript)

Secondly, beyond quantitative metrics, our qualitative content analysis of these policy texts reveals substantial differences in policy formulation approaches and content characteristics between the two phases. These distinctions enable clear categorization into distinct periods: the initial exploratory cultivation phase and the subsequent development and promotion phase. Our analysis identifies and examines representative policies from each period to illustrate these evolutionary changes. ( see lines 491-521 of the revised manuscript)

Thirdly, the comparative assessment of policy ratings between the two phases, supported by industry scale data from 2019 and 2022, provides empirical evidence of DPP's superior effectiveness in accelerating the growth of China's intelligent construction sector. This analysis substantiates the enhanced impact of DPP policies compared to their CEP predecessors. ( see lines 523-530 of the revised manuscript)

Finally,the transition from CEP to DPP reflects significant methodological advancements in policy formulation, manifested through three key dimensions ( see lines 535-562 of the revised manuscript):

1)The shift from experimental technology promotion to comprehensive industry application

2)Increased emphasis on long-term objectives and sustainable growth

3)Strategic responses to multiple driving factors

Factors Leading to the Policy Shift, Effectiveness of Phase 2 Policies, and Lessons for Future Policymaking

Our extensive discussion of policy text types, content themes, intergovernmental relationships, and the coupling of policy goals and tools in the original manuscript was specifically designed to examine the factors influencing policy transition and to identify the strengths and effectiveness of Phase 2 policies. To enhance the clarity and depth of this analysis, we have implemented the following modifications:

Firstly, we emphasize that the absence of any policy receiving a "perfect" rating indicates substantial room for optimization within CICP. Prior to proposing targeted improvement recommendations, it is essential to conduct an in-depth review and discussion o

---

## [Editor Report · Decision Letter 1]

Dear Dr. Ma,

Thank you for submitting your manuscript to PLOS ONE. After careful consideration, we feel that it has merit but does not fully meet PLOS ONE’s publication criteria as it currently stands. Therefore, we invite you to submit a revised version of the manuscript that addresses the points raised during the review process.

We look forward to receiving your revised manuscript.

Kind regards,

Ai Ren

Guest Editor

PLOS ONE

Journal Requirements:

Additional Editor Comments (if provided):

I would like to thank the authors for their thorough attention to all the comments and feedback provided. The revisions are well-considered, and I appreciate the effort put into addressing each point. While there are still a few minor issues with grammar and expression smoothing. For example, in the Unique Contributions session, change "This gap presents" to "These gaps present" to ensure consistency in plural form. Or, in the Relevance of Recommendations to the Research Findings session, change “expansion of policy nature" to "expansion of policy scope". Overall, I am satisfied with the revision and believe it has strengthened the manuscript. Thank you again for your hard work and dedication to improving the paper.

---

## [Author Response · Author response to Decision Letter 2]

21 May 2025

Dear Reviewer,

We sincerely appreciate your recognition of our revision efforts. We have carefully examined and polished the language and grammar in the paper. Please find below the implementation of your final suggestions:

Original Sentence/ Revision

Line25 conducts a comprehensive/ presents a systematic

Line30 reveal/ demonstrate

Line41 Utilizing The adoption of

Lines 45-46 These technologies are considered core because they significantly enhance construction efficiency and precision./ These core technologies significantly enhance construction efficiency and precision.

Line50 Together/ Collectively

Lines 56-57 policies at the national level, encouraging the development of the construction industry's informatization and intellectualization./ national policies to promote digitalization and intelligent transformation in the construction industry

Line66 Currently, research/ Research

Lines 67-69 A significant gap is the lack of quantitative evaluations of policy texts, which hinders the ability to assess their effectiveness with precision./ A critical research gap persists in the absence of rigorous quantitative evaluations of policy texts, significantly constraining the capacity to assess policy effectiveness with scientific precision.

Line106 covers/ spans

Line107 to propose/ aims to provide

Lines 125-127 Although these policy suggestions based on expert experience can reflect professionalism and forward-looking in some dimensions and perspectives, it is difficult to ensure the comprehensiveness and systematization of policy evaluation./ While expert-based policy suggestions reflect professionalism and foresight, ensuring their comprehensiveness and systematic evaluation remains a challenge.

Line132 made progress/ advanced

Line141 we aim/ this research aims

Line170 merges/ integrates

Lines 230-232 Methods that combine text encoding, data statistics, and graphical visualization were used in this research process, which could also be divided into working procedures, including Data Acquisition, Data Processing, Computation & Analysis, Results Discussion, and Conclusion./ This study employed methods such as text encoding, data analysis, and graphical visualization. The research process consisted of five stages: data acquisition, data processing, computation and analysis, results discussion, and conclusion.

Lines 243-244 the CICP evaluation using the PMC Index Model is the most critical section./ the evaluation of CICP using the PMC Index Model constitutes the most critical section.

Lines 267-268 The policy text (e.g., "Policy P0") is evaluated, and a standardized framework comprising ten primary variables is typically established, including "X1 Policy Nature," "X2 Policy Effectiveness," and so forth./ the policy text (e.g., 'Policy P0') is evaluated within a standardized framework of ten primary variables, including "X1 Policy Nature," "X2 Policy Effectiveness," and so forth.

Line274 Policy Nature/ Policy Scope

Line293 Policy Nature; policy nature/ Policy Scope; policy scope

Line381 Table 3. Research sample./ Table 3. Research sample of national intelligent construction policy texts (2010-2022).

Line400 Policy Nature/ Policy Scope

Line401 Policy Nature/ Policy Scope

Line438 , established in Table 5/ (Table 5)

Line451 "Acceptable" Class/ "Acceptable"

Line453 considering/ addressing

Line457 were deemed/ achieved

Line459 formulate/ formulation

Line486 a pivotal shift/ a transformative paradigm shift

Line517 evolutionary progression/ evolution

Line527 greater effectiveness/ stronger impact

Line572 Policy Nature/ Policy Scope

Line741 Effectiveness; goals/ efficacy; objectives

Line754 policy nature/ policy scope

Line770 Policy Nature/ Policy Scope

Line772 Policy Nature/ Policy Scope

Line774 Policy Nature/ Policy Scope

Line776 Policy Nature/ Policy Scope

Line783 policy nature/ policy scope

Line866 policy nature/ policy scope

Line882 comprehensive/ systematic

Line900 policy nature/ policy scope

Line916 Policy Nature Policy Scope

Line944 By implementing these recommendations, China can/ Implementing these recommendations would enable to

Thank you again for your time and comments. The suggestions from you are valuable for the advancement of this paper. We are happy to provide any further clarifications or revisions as necessary.

Sincerely,

Dr. Prof. Ming Ma

Institution: College of Architecture and Urban Planning, Yunnan University

Email: mming@ynu.edu.cn

TEL: 0086-18964833465

---

## [Editor Report · Decision Letter 2]

Research on quantitative evaluation of China's Intelligent Construction Policy (CICP) based on the Integration of PMC index model and multi-dimensional analytical framework

PONE-D-24-31927R2

Dear Dr. Ma,

We’re pleased to inform you that your manuscript has been judged scientifically suitable for publication and will be formally accepted for publication once it meets all outstanding technical requirements.

Kind regards,

Ai Ren

Guest Editor

PLOS ONE

Additional Editor Comments (optional):

Dear Authors,

Thank you for the opportunity to review the revised manuscript. I have carefully reviewed the responses to my comments and the revision.

All my concerns are addressed properly. The revisions have improved the clarity and quality of the manuscript, and I have no further suggestions for improvement.

Thanks

Best

Ai Ren
---

## [Editor Report · Acceptance letter]

PONE-D-24-31927R2

PLOS ONE

Dear Dr. Ma,

I'm pleased to inform you that your manuscript has been deemed suitable for publication in PLOS ONE. Congratulations! Your manuscript is now being handed over to our production team.

Kind regards,

on behalf of

Dr. Ai Ren

Guest Editor

PLOS ONE